# A Large-scale Dataset with Behavior, Attributes, and Content of Mobile Short-video Platform

## Abstract

Short-video platforms show an increasing impact on people's daily lives nowadays, with billions of active users spending plenty of time each day. The interactions between users and online platforms give rise to many scientific problems across computational social science and artificial intelligence. However, despite the rapid development of short-video platforms, currently there are serious shortcomings in existing relevant datasets on three aspects: inadequate user-video feedback, limited user attributes and lack of video content. To address these problems, we provide a large-scale dataset with rich user behavior, attributes and video content from a real mobile short-video platform. This dataset covers 10,000 voluntary users and 153,561 videos, and we conduct three-fold technical validations of the dataset. First, we verify the richness of the behavior data including interaction frequency and feedback distribution. Second, we validate the wide coverage of user-side and video-side attribute data. Third, we confirm the representing ability of the content features. We believe the dataset could support the broad research community, including but not limited to user modeling, social science, human behavior understanding, etc. Our dataset is available at this anonymous link: http://101.6.70.16:8080/[1].

## 1 Introduction

Short-video platforms such as Tiktok are redefining how users access information online. It is reported by Demand Sage[2] that Tiktok has attracted 1 billion active users who spend 1.5 hours each day on the platform on average. With the widely deployed algorithms of personalized recommendation, the platform can infer user preferences from users' behavioral logs and users' demographics (such as country, occupation, etc.), which helps alleviate information overload. However, the intelligence algorithm has also brought critical social concerns including echo chamber (Ge et al., 2020; Mosleh et al., 2021), filter bubble (Nguyen et al., 2014; Zhou et al., 2010), user addiction (Montag et al., 2018; Brand, 2022), etc. For example, it has been demonstrated that recommendation algorithms might lean to recommend popular items (Abdollahpouri & Mansoury, 2020) and this algorithmic bias could be amplified over time when users interact with recommender systems. The bias, in turn, sets constraints to the preference learning of algorithms, leading to the *feedback loop* (Mansoury et al., 2020), which further causes these concerns.

However, despite the fast growth of short-video platforms, there is no publicly-available dataset that well support the research of improving user modeling and alleviating the negative impact caused by AI algorithms. The existing public datasets of short-video platforms are released by researchers and engineers of recommendation algorithms, such as two of the largest short-video platforms, Kuaishou and WeChat Channel. However, they only focus on user feedback while are quite limited in supporting the research areas beyond recommendation algorithms. In specific, MicroVideo-1.7M (Chen et al., 2018) records millions of interactions between users and videos with 10,986 users and 1,704,880 for micro-video recommendation tasks. KuaiRec (Gao et al., 2022) is a fully observed micro-video dataset with additional user and video attributes collected from Kuaishou platform. REASONER (Chen

---

[1]Username: videodata Password: ShortVideo@10000
[2]https://www.demandsage.com/tiktok-user-statistics/

et al., 2023) is an explainable recommendation dataset, containing some basic attributes and ground truths for multiple explanation purposes. Tenrec (Yuan et al., 2022) is a large-scale multipurpose benchmark dataset for recommendation tasks, which also contains interactions between users and videos. MicroLens (Ni et al., 2023) is a recently released micro-video dataset with various modality content of videos. However, although these datasets may support the research of machine learning models for personalized user modeling or recommendation, they have not well covered the critical content such as video content and rich user attributes, leading to limited value for broader research, such as the study of filter bubble or user addiction. To sum up, they are suffering from the following limitations:

• **Lack of video content.** The video content is the most informative video-side data, which is neglected by most existing datasets, in which there is only coarse-grained categorical information of videos is provided.

• **Inadequate user-video feedback.** The existing datasets have limitations in the scale of the user set or item set, and the interaction number is limited. For example, REASONER (Yuan et al., 2022) covers only 2,997 users, 4,672 videos and 58,497 interactions. KuaiRec (Gao et al., 2022) includes only 1,411 users, 3,327 videos, and 4,676,570 interactions, in which user feedback is limited to play-finished and like.

• **Shortage of user attributes.** User attributes *i.e., demographics* are the basis of the research of understanding and further addressing the social-concerned issues in short-video platforms such as user addiction, however, these data are not sufficient in most existing datasets such as MicroVideo-1.7M (Chen et al., 2018), Tenrec (Yuan et al., 2022) and MicroLens (Ni et al., 2023).

In this paper, we provide a large-scale short-video dataset from 10,000 volunteers on one of the largest short-video platforms. The data collection procedure strictly follows privacy and ethical regulations (detailed in Section 2).

In terms of data scale, our dataset covers 1,019,568 interactions between 10,000 users and 153,561 videos, which is larger than existing released datasets, e.g., Kuairec (Gao et al., 2022) and REA-SONER (Chen et al., 2023). Besides, compared with datasets used in real industrial scenarios, our dataset has a similar scale to the datasets used to conduct offline experiments and analysis, such as the dataset of WeChat Channel [3] (7,210,290 interactions of 20,000 users with 96,428 short videos) and Tiktok [4] (1,047,358 interactions of 16,538 users with 366,017 short videos). Therefore, our dataset can satisfy the requirements of both academic and industrial scenarios.

In terms of data coverage, our dataset has wider coverage than existing public datasets in three aspects: user behavior, user/video attributes and video content. The user behavior data includes user's historical watching records (user ID, video ID, interaction timestamp), six kinds of explicit feedback conveying obvious preference (including like, follow, forward, collect, comment and hate) and implicit feedback indicated by watching time which indirectly reflects user satisfaction. As for attribute data, we collect rich side information of users and videos in which most are overlooked by previous datasets. User-side attributes contain three categories: demographic attributes (*e.g.* age, gender), geographical attributes (*e.g.* city, community type) and some other attributes (*e.g.* phone price). Video-side attributes cover three-level content categories, author information (*e.g.* author ID, number of fans), title text and tag. By comparison, previous datasets (Chen et al., 2018; Li et al., 2019) only cover a subset, *e.g.* age, gender and single-level categories. In terms of content data, our dataset is the first to provide the raw video files which are always unavailable in existing datasets but essential for describing and understanding video content. Moreover, we provide extracted visual features and ASR results of videos for convenient use. We further conduct the analysis of the basic characteristics of behavior and attribute data, verifying the comprehensiveness and distribution evenness of different data fields. We also validate the quality of the content data through an embedding visualization method and observe obvious clusters between different video categories. Finally, we discuss how our dataset benefits the broad research community from three aspects: user modeling, social science, and human behavior understanding. We summarize our contributions as follows:

• We collect a large-scale dataset from a real mobile short-video platform, covering rich user behavior, attributes and video content, which are scarce in existing datasets.

---

[3]https://algo.weixin.qq.com/2021/problem-description
[4]https://www.biendata.xyz/competition/icmechallenge2019

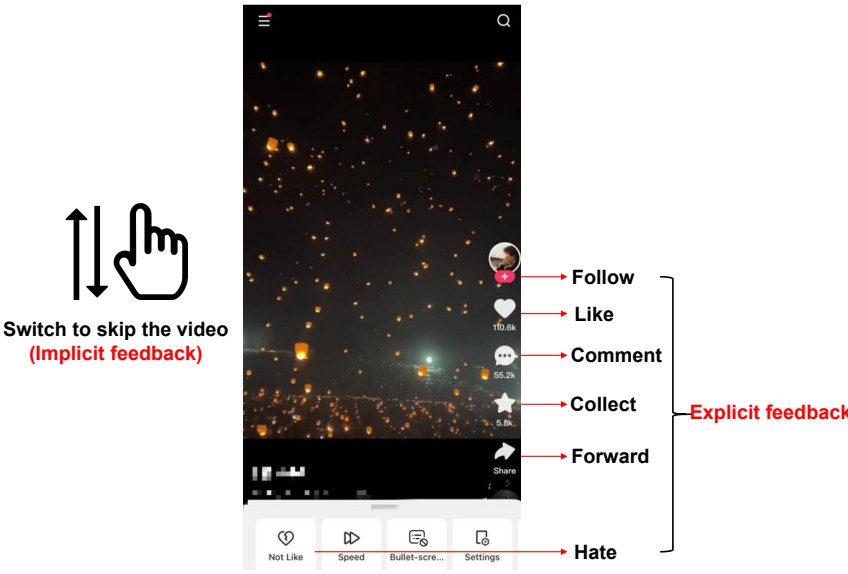

Figure 1: Illustration of the user interface and behaviors on the platform.

- We conduct comprehensive technical validation of the wide coverage of behavior and attribute data and the quality of content data in the dataset.
- We provide a sufficient discussion of the potential research directions with our dataset, covering broad research communities such as user modeling, social science, and human behavior understanding.

## 2 DATA DESCRIPTION

Our dataset possesses much richer information than existing datasets (Chen et al., 2018; Gao et al., 2022; Li et al., 2019) on three aspects. First, the dataset contains large-scale interaction records and multi-type user behavior helpful for user interest modeling and behavior analysis. Second, the dataset contains various user attribute data which has never been covered by existing datasets. Third, the video content is available in the dataset, which could support further information extraction and analysis. An overview of the dataset is shown in Figure 2. The structure and detailed data description are presented in the supplementary material.

### 2.1 BEHAVIOR DATA

We collected six-month historical user interactions of the 10,000 volunteers (the data of users under 20 years old has been removed from the actual dataset) under their consent, while in the paper we focus on analyzing the first week's data for a quick release. There are 153,561 involved videos and 1,019,568 interaction records. For each interaction record, we record the basic information including anonymous user/video ID and timestamp of the interaction. More importantly, we have collected rich user behavior which could convey diverse feedback signals of users. On the platform we use, users can watch a variety of short-form videos, which are organized by recommending streaming where each time the user can see only one video, which is known as the single-column user interface. In this scenario, the video will automatically play, during which users can choose to stop viewing the current video by exiting this interface or switching to the next video. Overall there are mainly two types of feedback in this scenario: implicit feedback and explicit feedback.

Implicit feedback refers to the user's watching behavior, more specifically, the watching time, which is a vital signal to yield user satisfaction (Zheng et al., 2022; Liu et al., 2021). Traditional datasets usually focus on discrete user feedback, such as ratings (Koren, 2008; Koren et al., 2009) and click

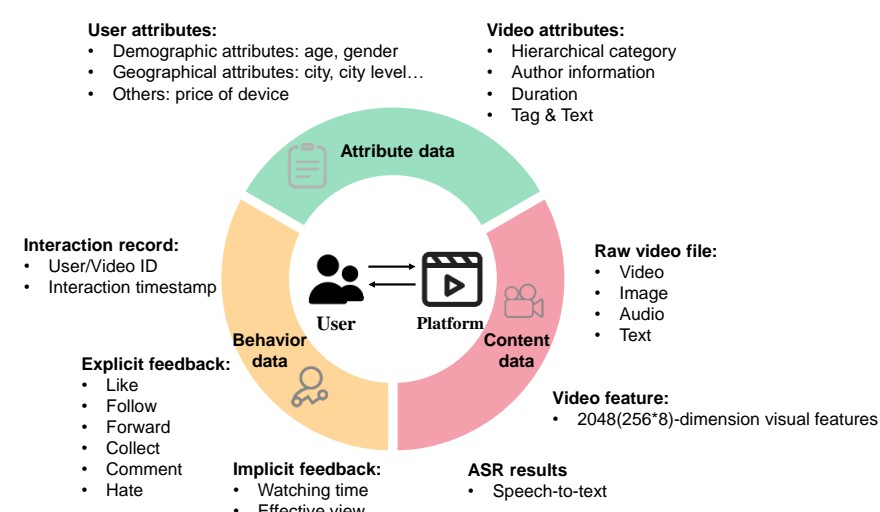

Figure 2: An overview of the dataset, which contains behavior data, attribute data and content data.

behavior (Liu et al., 2020; Song et al., 2019). Differently, in the scenario of short-video platforms, users will watch the current video recurrently until they switch to the next video as Figure 1 shows, in which case user engagement is mainly reflected on the continuous watching time (Salganik et al., 2006). Intuitively, users tend to continue watching if they are interested in the current video, otherwise, they will skip to the next video. Put another way, the watching time could serve as a critical indicator of user interest. The behavior data not only records the precise watching time of each interaction but also provides an "effective view" label by comparing watching time and a threshold (set as 3 seconds in our dataset).

Explicit feedback, provided by users actively, usually conveys clear preference or disgust. For example, if the user collects or forwards a video, we could infer that they are interested in the video content with high confidence. Conversely, if a user conveys hate to the video, it could be assumed reasonably that they show no interest in it. The behavior data records 6 kinds of explicit feedback including like, comment, follow, forward, collect and hate.

## 2.2 ATTRIBUTE DATA

Compared with existing datasets, the dataset contains richer attribute information about the users and videos. Specifically, there are 9 video-side attributes and 6 user-side attributes in the dataset, shown in the supplementary material.

For the video side, we have fully exploited the key attributes of micro-videos including video category, author information, duration, video text and video tag.

- **Video category**. According to the regulation of the short-video platform, video authors are required to provide the title and tag for uploaded videos for better management and recommendation. We establish the hierarchical categories by integrating the semantics of the title and tag, which can cover most of the collected videos. As for the videos hard to categorize by the title and tag, we train a classification model based on labeled data and take the prediction as the categories of these videos.

  The category of videos is divided into 3 levels hierarchically. For each video, the primary, secondary, and tertiary category is represented by "Category I", "Category II" and "Category III", respectively. For example, for a video recording a *hockey game*, "Category I" is "*sports*", "Category II" is "*ball game*" and "Category III" is "*hockey*". Statistically there are 37 kinds of "Category I", 281 kinds of "Category II" and 382 kinds of "Category III". It is worth mentioning that the number of Category II and Category III are similar since some Category II could be further divided into several Category III while some others could not. The hierarchical category organization possesses

richer semantic information compared with existing datasets, which could support multi-level video content understanding (Diba et al., 2020). In fact, both coarse-grained and fine-grained categories are indispensable. For example, in video advertising, the coarse-grained category is more appropriate to enlarge the audience group. While in tasks such as video retrieval, fine-grained categories are necessary to ensure retrieval accuracy. Besides, the relation between different categories is valuable, for example, if a user likes videos about delicious food, the platform could reasonably recommend videos about cooking, improving the diversity of the recommendation list.

- **Author information**. The dataset totally covers 81,870 video authors in all. We collect the author information including author ID and the number of fans, which is vital to the video-side modeling and extends the social relation information on the platform. On a User-Generated Content (UGC) platform like short-video platforms, author information (*e.g.* the number of fans) is useful for social science research, such as studying social capital of short-video authors including social networks and the recognition on the platform (Zhuge, 2018). This information is also indispensable in research of popularity bias (Tang et al., 2022) and celebrity effect(Luo et al., 2021).

- **Duration**. The dataset covers videos with diverse duration, varying from less than 30 seconds to larger than 5 minutes and the majority are relatively short (less than one minute). In the research of user addiction and engagement, it's necessary to take the duration of watched videos into consideration. For example, some research (Yang et al., 2021) has studied watching behaviors on the short-video platform and found videos with less duration tend to be more addictive.

- **Video text**. The text of the video title is available in our dataset while often overlooked in existing datasets. It summarizes the semantic content of videos and enhances video content understanding (Zolfaghari et al., 2018; Du et al., 2018). There are 140,341 unique titles in all videos.

- **Video tag**. Video tag is added manually by authors shown in each video for better video description and understanding, such as "*delicious food recommendation*" and "*travel tips*". Note that video tags are collected from the platform while categories are processed and classified through our human effort. In practice, it has been utilized to summarize video content and support video understanding (Shamma et al., 2007; Mazeika et al., 2022). There are a total of 79,705 unique tags in all collected videos.

For the user side, the dataset records comprehensive user attributes including demographical, and geographical characteristics, which are not adequate in existing datasets. This part of data is optionally provided by volunteers and not involved with privacy issues since the user ID for each volunteer has been hashed for privacy protection.

- **Demographic attributes**. The demographic attributes include gender and age of users, which are basic for the research of social-concerned issues in short-video platforms like user addiction (Montag et al., 2018; Brand, 2022). For example, in the research on the fairness of recommender systems (Geyik et al., 2019; Wu et al., 2022c), gender and age are vital sensitive attributes for the selection of protected user groups. In the data analysis of the paper, we do not exclude the minor users but their data has been removed in the actual dataset.

- **Geographical attributes**. The geographical attributes here contain the city each user lives in, the city's level (first-tier, second-tier, third-tier, fourth-tier, fifth-tier) and the community type (country, urban area and town). The geographical characteristics could complement the user-side profile and support the analysis of regional user differences.

- **Other attributes**. Our dataset records the price of user devices, which is seldom covered by previous datasets (Chen et al., 2018; Gao et al., 2022). Considering privacy protection, we only collect the phone's model name with the permission of volunteers and retrieve the corresponding phone price.

## 2.3 CONTENT DATA

The video content (raw video files, processed content features and ASR results) is the most prominent highlight of our dataset, which has always been neglected by existing public datasets. Our dataset fills in this blank by collecting the raw file of all 153,561 videos that appeared in the interaction records. The total duration of all videos reached 3,998 hours and the total file size is 3.2 TB. On the basis of raw video files in our dataset, for convenience of use, we have also conducted some preprocessing

| Model | Recall@10 | Recall@20 | Recall@50 | NDCG@10 | NDCG@20 | NDCG@50 |
|---|---|---|---|---|---|---|
| BPR | 0.0113 | 0.0218 | 0.0453 | 0.0078 | 0.0114 | 0.0179 |
| LightGCN | 0.0223 | 0.0358 | 0.0629 | 0.0169 | 0.0211 | 0.0286 |
| LayerGCN | 0.0208 | 0.0336 | 0.0559 | 0.0160 | 0.0198 | 0.0260 |
| VBPR | 0.0184 | 0.0283 | 0.0519 | 0.0140 | 0.0172 | 0.0237 |
| MMGCN | 0.0105 | 0.0187 | 0.0330 | 0.0088 | 0.0114 | 0.0155 |
| GRCN | 0.0119 | 0.0224 | 0.0444 | 0.0089 | 0.0125 | 0.0189 |
| BM3 | **0.0238** | **0.0364** | **0.0638** | **0.0178** | **0.0218** | **0.0294** |
| LGMRec | 0.0159 | 0.0257 | 0.0449 | 0.0131 | 0.0162 | 0.0217 |

Table 1: Benchmark results on 8 recommendation algorithms (including both general and multimodal recommendation methods) to validate the practical usage of our dataset on user modeling.

towards the raw videos. Specifically, we first divide each video into 8 clips uniformly according to its length, then extract the 256-dimension visual feature of each clip through a pre-trained convolutional neural network(here we use Resnet (He et al., 2016) architecture). In addition, we also provide bilingual (Chinese and English) ASR text extracted with SenseVoice-Small (SpeechTeam, 2024). The detailed description of content data is shown in the supplementary material.

## 3 TECHNICAL VALIDATION

In order to ensure the technical validity of the collected data, we have conducted validation from four aspects. First, we establish a benchmark of recommendation algorithms using our dataset. Second, we verify the richness and diversity of the behavior data including interaction frequency distribution and feedback distribution. Third, we validate the coverage of different fields for both user-side and video-side attribute data. Finally, we carry out the quality validation of the content data, especially the preprocessed visual features of videos through the dimension reduction and visualization procedure.

### 3.1 BENCHMARKING RECOMMENDATION ALGORITHMS

We provide experimental results of 8 recommendation algorithms in Table 1, to validate the practical usage of our dataset on user modeling. The tested methods can be categorized into general and multimodal recommendation methods. The general recommendation method includes BPR (Rendle et al., 2009), LightGCN (He et al., 2020) and LayerGCN (Zhou et al., 2023b). The multimodal recommendation method includes VBPR (He & McAuley, 2016), MMGCN (Wei et al., 2019), GRCN (Wei et al., 2020), BM3 (Zhou et al., 2023c) and LGMRec (Guo et al., 2024). We divide the train, validation and test set with the ratio 8:1:1, following the process pipeline introduced in (Zhou et al., 2023a). From the result, BM3 achieves the best performance in the current compared methods by effectively utilizing multimodal information, which is similar to the observations on other multimodal recommendation benchmarks (Zhou et al., 2023a). This demonstrates the practical usage of our dataset for benchmarking recommendation algorithms.

### 3.2 COVERAGE OF BEHAVIOR DATA

In the validation of the richness and diversity of behavior data, we conduct a two-fold analysis: the interaction frequency distribution, and the distribution of explicit and implicit feedback.

#### 3.2.1 DISTRIBUTION OF INTERACTION FREQUENCY

We first divide the users and videos according to their interaction frequency, the distribution of which is shown in Figure 3. It can be seen that overall the dataset covers users with diverse activities and videos with diverse exposure. The user interaction number ranges from 1 to 2,228. The most common interaction number lies in 0-30, occupying 36.66% of all volunteers. Overall, the user number decreases with the increasing interaction frequency as presented in Figure 3a, consistent with the fact that extremely active users account for only a small portion. In terms of the videos, according to the distribution shown in Figure 3b, there are sundry videos with different numbers of interactions, from 1 to 1,193. The videos with 0-5 interactions are the most common (accounting for

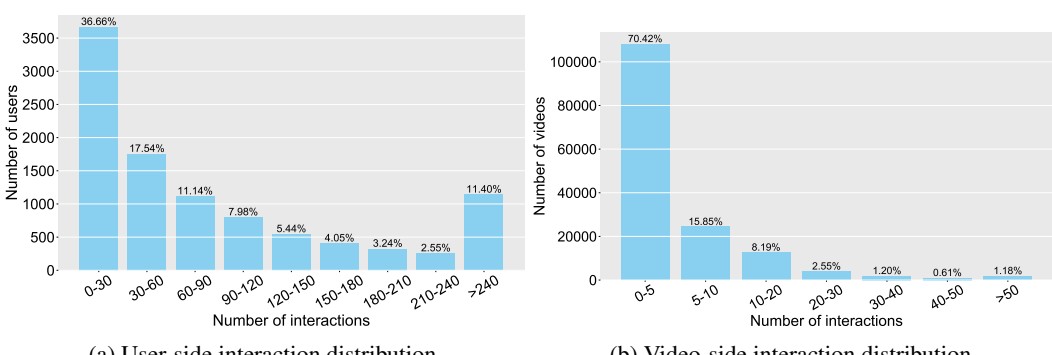

(a) User-side interaction distribution.

(b) Video-side interaction distribution.

Figure 3: Interaction number distribution of (a) users and (b) videos.

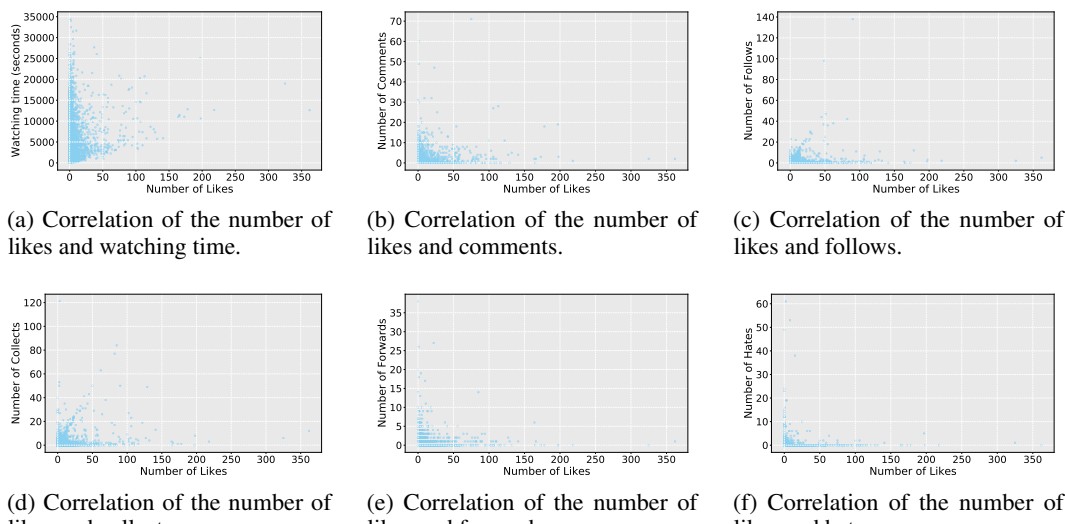

(a) Correlation of the number of likes and watching time.

(b) Correlation of the number of likes and comments.

(c) Correlation of the number of likes and follows.

(d) Correlation of the number of likes and collects.

(e) Correlation of the number of likes and forwards.

(f) Correlation of the number of likes and hates.

Figure 4: Correlation of the number of likes and the number of (a) watching time, (b) comments,(c) follows, (d) collects, (e) forwards, (f) hates per user.

70.42%) and only 1.18% videos are watched over 100 times since the platform controls the exposure distribution of videos and only a few high-quality videos could get broad exposures. To sum up, the dataset covers diverse users and videos in terms of interaction frequency.

### 3.2.2 EXPLICIT AND IMPLICIT FEEDBACK DISTRIBUTION

The user behavior on the platform is divided into two parts: explicit feedback and implicit feedback. Explicit feedback such as like, comment and hate on the platform is sparse. By comparison, implicit feedback (*i.e.,* continuous watching behavior) is much more common. The distribution of user-side and video-side watching time is shown in the supplementary material, from which it can be seen that most volunteers watched within 1,000 seconds in the 7 days, and the videos that have been watched for less than 50 seconds are the most common.

Furthermore, aiming to get a deeper understanding of user behavior habits, we conduct a detailed analysis of the relation between explicit and implicit feedback by calculating Pearson correlation and the p-value between them, and the results are shown in Figure 4. In this part, the number of explicit feedback is counted during the whole data collection process (i.e., the accumulated number on the seventh day). From the result, the following conclusions can be derived:

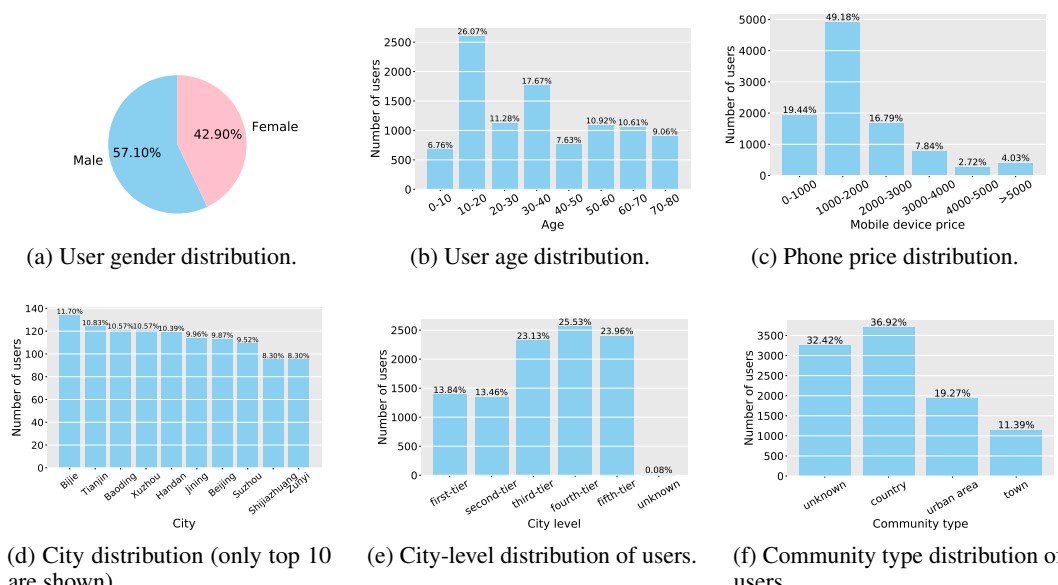

(a) User gender distribution.

(b) User age distribution.

(c) Phone price distribution.

(d) City distribution (only top 10 are shown).

(e) City-level distribution of users.

(f) Community type distribution of users.

Figure 5: Distribution of the six fields in user attributes.

- **Watching time and like behavior have weak positive relations with statistical significance.** The Pearson correlation is 0.20 (significant at the 0.001 level). The users providing more like behaviors usually have a long watching time. The reason might be that these active users usually watch more videos, thus having more opportunities to find satisfying videos on the platform.

- **Like behavior and follow/collect/comment behavior have weak positive relations with statistical significance.** The Pearson correlation with follow/collect/comment are 0.18 (significant at the 0.001 level), 0.19 (significant at the 0.001 level) and 0.09 (significant at the 0.01 level). The users with more like behavior also show higher frequency in follow and collect behavior. It is reasonable because these behaviors all convey user interest and usually happen simultaneously.

- **Like behavior and forward/hate behavior have no statistically significant relations.** The Pearson correlation between the number of likes and forwards/hates are 0.04 and -0.03 but both are not statistically significant.

### 3.3 COVERAGE OF ATTRIBUTE DATA

In order to validate the coverage of the attribute data, we have derived the distribution of the key information related to users and videos. For the user side, there are mainly two kinds of attributes: demographic attributes (including gender and age), and geographical attributes (including city, community type and city level). We show the distribution of each attribute in Figure 5.

- **Distribution of demographic attributes.** The demographic attributes cover the gender and age of users in the dataset. In all volunteers, the ratio of male and female users is 57.1% and 42.9% respectively, which exhibits a relatively uniform distribution. The user age ranges from 5 to 79 and the 10-20 year-old users occupy the largest proportion (26.07%) showing that young users are important user groups of the platform. According to the official report, there were users with 44% female and 56% male, which is highly consistent with the distribution in our dataset without obvious bias. This provides evidence for the reliability of the overall collected data.

- **Distribution of geographical attributes.** As for the geographical attributes, the dataset covers 373 cities (we only show the top 10 due to the limit of space), 3 community types(country, urban area, town) and 5 city levels (first-tier, second-tier, third-tier, fourth-tier and fifth-tier). In this dataset, most users come from the country, occupying 36.92%. The users from urban areas and towns are 19.27% and 11.39%, respectively. 32.43% of users don't provide this information and we label their community type as "unknown".

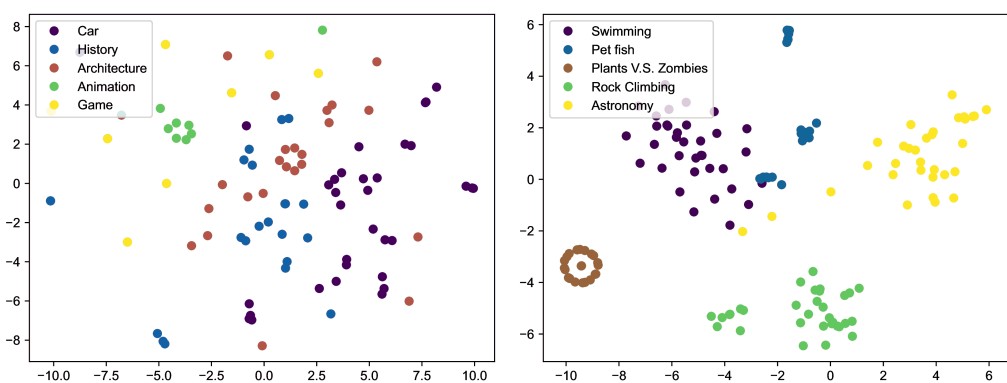

(a) Embedding visualization across five Category I.  (b) Embedding visualization across five Category III.

Figure 6: Embedding visualization of videos with different (a) Category I and (b) Category III through t-SNE.

For the video side, we focus on the validation of two important attributes: category and duration.

- **Video category.** Overall, the dataset covers 37 primary categories(Category I), 281 secondary categories(Category II) and 382 tertiary categories(Category III). The distribution of Category I is presented in the supplementary material.

- **Duration.** The dataset includes videos with a wide range of duration from 2.7 to 1734.9 seconds, and the distribution of the duration is shown in the supplementary material. In the dataset, most videos are relatively short with less than 30 seconds. Overall, the number of videos decreases with the increasing duration, which is consistent with the concept of short-video platforms.

In summary, the dataset includes rich attribute data, which largely extends the range of user and video information compared with existing datasets. On the whole, the distribution of the attributes is comparatively even which could reduce the analysis bias.

### 3.4 Quality Validation of Video Content Data

One significant difference between our dataset and existing datasets is that we provide raw video files and preprocessed visual features which could support better utilization in research. To examine the quality of the extracted visual features in a comprehensible way, we first conduct an embedding visualization using t-SNE (Van der Maaten & Hinton, 2008), which is widely used for dimension reduction and suitable for visualizing high-dimension data. Specifically, we randomly pick five categories of videos from the dataset on Category I-level and Category III-level, respectively. The result is presented in Figure 6, from which it can be seen that videos with different categories have discriminative content features. This validates that the preprocessed content features have efficiently encoded semantic information of the videos, which can well support the further study of video understanding, video sentiment analysis and user interest modeling. For example, the visual features could serve as a valuable tool for analyzing the sentiment conveyed by the video, providing insights for measuring the emotional dynamics of video generators.

We then validate the practical usage of extracted video features via a downstream recommendation task. We test the performance of 4 typical multimodal recommendation methods with and without video features, as shown in Table 5 of the Appendix. From the results, it can be seen that there's a common performance drop without video features of all methods, demonstrating the value of extracted video features.

## 4 Potential Research Directions with the Dataset

We discuss the intended research directions that our dataset could support as follows:

- **User modeling and personalization.** Nowadays making accurate user modeling and personalized recommendations has become one of the main focuses of academia and industry (Wu et al., 2022a;b; Volkovs et al., 2017). On the short-video platform, introducing rich multimodal information from videos can help achieve better user modeling and personalization. However, these data are not well supported by existing datasets due to the shortage of some necessary information such as video content. By comparison, our dataset covers broader content information of users and videos compared with existing datasets. For example, the available raw video files could support more fine-grained multi-modal feature extraction and enhance content-based recommendation.

- **Fairness of AI algorithms.** It has long been blamed that artificial intelligence algorithms might lead to and amplify inequity and unfairness (Kusner & Loftus, 2020; Espín-Noboa et al., 2022). One important concern with short-video recommender systems is the potential for bias or discrimination towards different users or video groups. For the user side, recommender systems might exhibit unintentional discrimination across groups with different genders or ages. For the video side, if the recommender system favors certain categories of content or authors, it might lead to unequal exposure for different groups of videos. Our dataset covers necessary user-side (*e.g.* gender, age, living city) and video-side information (*e.g.* hierarchical categories and raw video content) for multi-aspect group division, which could support the research on the fairness issue.

- **Filter bubble on micro-video platforms.** The filter bubble phenomenon has attracted increasing attention in recent years, which indicates algorithms used by online platforms could cause a decrease in information diversity (Zhou et al., 2010; Bozdag & Van Den Hoven, 2015). The recommendation algorithms tend to make recommendations based on users' historical preferences, thus reinforcing their existing interests and limiting exposure to diverse content, which harms user experience and platform development. The filter bubble refers to the situation where recommendation algorithms overly tailor content to a user's preference, resulting in limited content exposure for the user on the platform. To quantify content coverage, our dataset includes a hierarchical content category system. Coverage at a category level c is defined as $\frac{N_{seen}(u,c)}{N_{all}(c)}$, where $N_{seen}(u,c)$ represents the number of level-c categories a user $u$ has interacted with over a period, and $N_{all}(c)$ is the total number of level-c categories available. A user's filter bubble extent can then be determined by comparing this metric against a threshold, such as the average across all users. Subsequent analyses could investigate how user behaviors, such as viewing time and content preferences, influence the severity of filter bubbles across different category levels. Additionally, tracking the temporal evolution of filter bubbles at each level could provide insights into their dynamics and persistence over time.

- **Polarization on online platforms.** The polarization phenomenon on social networking platforms might occur when users are exposed to limited content that reinforces their existing beliefs (Sikder et al., 2020; Santos et al., 2021). One potential way to analyze the polarization phenomenon is to identify clusters of users who are exposed to similar content and show similar attitudes towards the same video. Our dataset covers various kinds of feedback representing their clear beliefs and interests, which could support the identification and analysis of the polarization phenomenon on micro-video platforms.

- **User addiction on micro-video platforms.** User addiction on the Internet is a growing concern nowadays (Montag et al., 2018; Brand, 2022). Analysis of user addiction relies on the tracking of time spent on the platform and the consumed content. Our dataset covers large-scale records of users' watching history on the micro-video platform, together with an accurate timestamp and content information, which can be used to conduct analysis on the user addiction problem.

## 5 CONCLUSIONS

In this paper, we introduce a large-scale dataset with rich user behavior, attributes and video content from a real mobile short-video platform, where we detail the data collection process, data characteristics and quality validation. Our dataset can support broad research communities such as user modeling, social science, human behavior understanding and so on. We have released the whole dataset and codes for data character analysis to facilitate relevant research. In the future, we plan to provide more fine-grained video content like semantic objects and sentiment labels so as to better describe video content. Besides, we would like to process more data and provide user interactions with longer periods on the platform.

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

## A  SUPPLEMENTARY MATERIAL

### A.1  DATA COLLECTION

In this section, we describe how we collect the datasets, addressing the shortcomings of the existing publicly-available datasets. Given the limitation of existing datasets, we try to build a dataset with behaviors, attributes, and content. Since the user-side data is highly sensitive to user privacy, we hire volunteers to obtain their interaction with videos, along with attributes. It's worth noting that our data is collected from the consenting volunteers. Considering that the volunteer group is large enough and covers a wide range of user attributes, the representativeness of the experimental group can be guaranteed. Specifically, we asked the volunteers to install a proxy agent on their mobile devices and sign up for a new account on one of the largest short-video platforms. The proxy agent can collect the users' behaviors on the short-video app, and all volunteers fully acknowledge how the proxy agent works. Volunteers are asked to keep the agent installed for one week, and we are then able to collect enough behavioral logs. During the data collection process, the volunteers are requested to freely interact with the videos, following their own interests, without any prior restrictions. That is, we either encourage or discourage volunteers from using the app. With the proxy agent, we can collect (and only collect) the behaviors on the short-video App, including various user-video feedback, such as watch, click, like, share, etc. The proxy also has access to the watched videos cached in the memory of devices, making it possible to collect the video content. There are mainly four components in the data collection process with the proxy agent:

- **Traffic interceptor.** In order to collect the traffic between clients and the short-video platform, we use MITMproxy to transparently intercept all HTTP/HTTPS traffic between the mobile device and the platform. With the MITMproxy CA certificate installed on the mobile device, the proxy can collect data including the cached videos, video attributes such as video titles, author information and behavior information, *e.g.* user/video ID and request timestamps.

- **Video collection.** The proxy can access the cached videos that have been watched by users and we export these files as video content data.

- **Screen-touching listening.** The proxy agent can also automatically capture the screen-touching behavior backstage including click, slide and long press, etc. With the collected event, timestamp and position, we can infer the feedback provided by volunteers when watching videos such as like, collect and hate.

- **Privacy protection.** Considering the need for privacy protection, we conducted hashing towards all user/video/author IDs and dropped all sensitive information during data collection. Besides, We conducted some modifications and summarization on the video title to make it hard to identify but reserved the keywords and original meaning.

The above four designs of the proxy agent are transparent and all get the agreement of volunteers. Furthermore, the volunteers are required to fill an optional questionnaire for the basic demographics. With the above operations, we collect the dataset with behaviors, video content, and user attributes. The whole data collection procedure is illustrated in Figure 7. To ensure the quality of collected data, we have the following actions:

- **Content filtering.** We conducted a further check and removed the offensive videos from our dataset. These operations ensure that the offensive content has been excluded from our dataset.

- **Self-reporting information check.** To mitigate the self-reporting biases of volunteers, we made two-fold efforts. On the one hand, we introduced some special designs in the questionnaire used for collecting user-side information, which can help assess the consistency and validity of responses. For example, we have designed questions that ask respondents to convey their attributes in different ways. For the answers with conflicts, we have filtered out these volunteers in the dataset to reduce self-reporting bias. On the other hand, we combined the self-reported data with other sources of information (e.g., comparing the IP address of users recorded by the platform and self-reporting cities) to cross-validate responses and identify potential discrepancies.

*Ethical Statements.* The data collection was conducted strictly following the local ethical regulations which have similar privacy regulations as GDPR and was supervised by the Science and Technology Ethics Committee. Before the submission, the dataset has passed the data assessment of our committee,

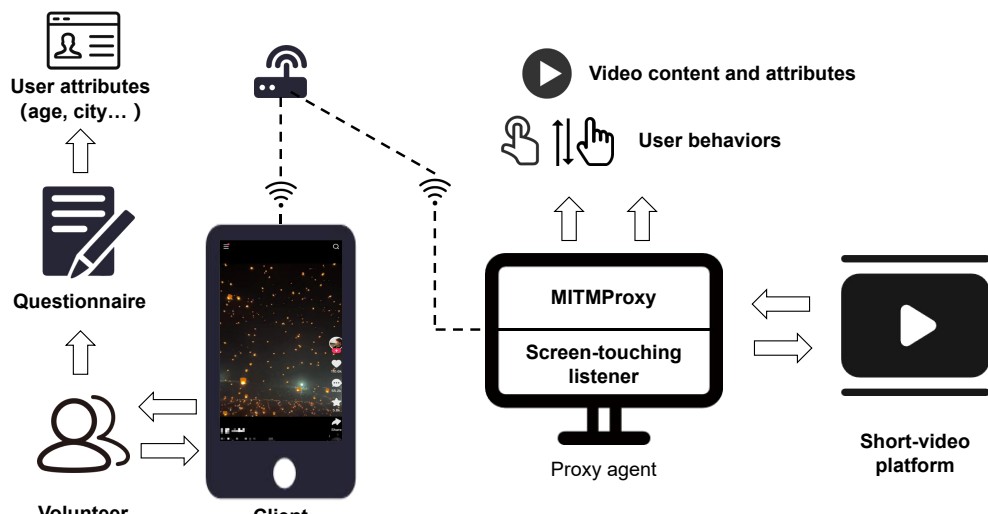

Figure 7: Illustration of the whole data collection process, including video content and behavior data collection with the proxy agent and user attributes collection with questionnaires.

including the check of data content, collection procedure, collection purpose, privacy and safety guarantee. For each participant, we collected seven pieces of information: user ID, gender, age, device price, name of the residential city, city's level, and community type. The user ID has been hashed, which was mapped to a number between 1 and 10,000 to prevent the identification of users. For other demographic and geographic attributes, we only collected coarse-grained data, for example, the geographic granularity is limited to the city level. Such information is not sufficient to be utilized for the identification of users. Additionally, all the participants have signed the informed consent form and agreed their anonymized information to be used for research purposes. Besides, they have the right to opt out at any time. Once the volunteer proposes to opt out in the future, we will update the dataset timely to erase all relevant data of the volunteer. Besides, we will inform the community on the data-sharing website that they should download and use the newest version of the dataset for privacy protection.

## A.2 DATASET DOCUMENTATION AND INTENDED USE

### A.2.1 DATASHEET

We utilize the datasheet format (Gebru et al., 2021) for dataset documentation.

*Motivation:*

- **For what purpose was the dataset created?** Existing short-video datasets have not well covered the critical content such as video content and rich user attributes, leading to limited value for broader research. The dataset is created to facilitate broad research directions such as user modeling, social science, and human behavior understanding.

- **Who created the dataset (e.g., which team, research group) and on behalf of which entity (e.g., company, institution, organization)?** The paper authors.

- **Who funded the creation of the dataset?** This work is supported in part by the National Natural Science Foundation of China and the National Key Research and Development Program of China.

- **Any other comments?** No.

*Composition:*

- **What do the instances that comprise the dataset represent (e.g., documents, photos, people, countries)?** The dataset includes users and videos on the short-video platform.

- **How many instances are there in total (of each type, if appropriate)?** This dataset covers 10,000 users and 153,561 videos.

- **Does the dataset contain all possible instances or is it a sample (not necessarily random) of instances from a larger set?** It is a sample from all users and videos from the short-video platform. We have verified the coverage and representativeness of the dataset in the section "Technical Validation".

- **What data does each instance consist of?** The behavior and attribute data are collected for users. The attribute data and raw files are collected for videos. Detailed descriptions are shown in Table 2, 3 and 4.

- **Is there a label or target associated with each instance?** No.

- **Is any information missing from individual instances?** No.

- **Are relationships between individual instances made explicit (e.g., users' movie ratings, social network links)?** There are explicit relationships between users and videos according to user behavior on the platform such as watching, like, collect, forward, hate, etc.

- **Are there recommended data splits (e.g., training, development/validation, testing)?** Data can be split according to the interaction time between users and videos.

- **Are there any errors, sources of noise, or redundancies in the dataset?** There might be some self-reporting errors. To address this issue, we have introduced some special designs in the questionnaire used for collecting user-side information, which can help assess the consistency and validity of responses. For example, we have designed questions that ask respondents to convey their attributes in different ways. For the answers existing conflicts, we have filtered out these volunteers in the dataset to reduce self-reporting bias. Besides, we have combined the self-reported data with other sources of information (e.g., comparing the IP address of users recorded by the platform and self-reporting cities) to cross-validate responses and identify potential discrepancies.

- **Is the dataset self-contained, or does it link to or otherwise rely on external resources (e.g., websites, tweets, other datasets)?** The dataset is self-contained.

- **Does the dataset contain data that might be considered confidential (e.g., data that is protected by legal privilege or by doctor-patient confidentiality, data that includes the content of individuals' non-public communications)?** No.

- **Does the dataset contain data that, if viewed directly, might be offensive, insulting, threatening, or might otherwise cause anxiety?** No.

- **Does the dataset relate to people?** Yes, the dataset contains anonymous information of voluntary users, fully under their consent.

- **Does the dataset identify any subpopulations (e.g., by age, gender)?** Yes, the attribute data of users covers age and gender data. The distribution is validated to be balanced in the section of "Technical Validation".

- **Is it possible to identify individuals (i.e., one or more natural persons), either directly or indirectly (i.e., in combination with other data) from the dataset?** No. We have anonymized IDs of users and videos. Besides, we have discretized the video duration field into bins and conducted some modifications and summarization on the video title to make it hard to identify.

- **Does the dataset contain data that might be considered sensitive in any way (e.g., data that reveals racial or ethnic origins, sexual orientations, religious beliefs, political opinions or union memberships, or locations; financial or health data; biometric or genetic data; forms of government identification, such as social security numbers; criminal history)?** No.

- **Any other comments?** No.

*Collection process:*

- How was the data associated with each instance acquired? We collect the dataset from voluntary users and their interacted videos, detailed in Section A.1.

- **What mechanisms or procedures were used to collect the data (e.g., hardware apparatus or sensor, manual human curation, software program, software API)?** The user attribute data is collected from the survey. The video attribute and content data and the user behavior data are collected from software programs.

- **If the dataset is a sample from a larger set, what was the sampling strategy (e.g., deterministic, probabilistic with specific sampling probabilities)?** During the data collection, we sample users to keep the distribution of critical attributes nearly balanced, e.g., gender.

- **Who was involved in the data collection process (e.g., students, crowdworkers, contractors) and how were they compensated (e.g., how much were crowdworkers paid)?** In total, we recruit 10, 000 voluntary users of the short-video platform, and we pay each $6 for completing the data collection.

- **Over what timeframe was the data collected?** From September 16, 2022 to September 22, 2022.

- **Were any ethical review processes conducted (e.g., by an institutional review board)?** The dataset has passed the data assessment of our administration, including the check of data content, collection procedure, collection purpose, privacy and safety guarantee.

- **Does the dataset relate to people?** Yes, the data collection involves hired volunteers.

- **Did you collect the data from the individuals in question directly, or obtain it via third parties or other sources (e.g., websites)?** From the individual survey for each volunteer.

- **Were the individuals in question notified about the data collection?** Yes, we have taken great care in informing volunteers about the data collection process and the purpose of the study. More importantly, we have informed volunteers that their responses will be kept anonymous and have no influence on their daily lives.

- **Did the individuals in question consent to the collection and use of their data?** Yes, we collect the data fully under the consent of volunteers. Specifically, the collection procedure, collection purpose, privacy and safety guarantee have been listed in the informed consent which all volunteers have read carefully before data collection.

- **If consent was obtained, were the consenting individuals provided with a mechanism to revoke their consent in the future or for certain uses?** Yes, at the beginning of the data collection, we took great care in informing all of the volunteers that their data would be anonymized to be unidentified for privacy protection and they have the right to opt-out at any time. Once the volunteer proposes to opt out in the future, we will update the dataset timely to erase all relevant data of the volunteer. Besides, we will inform the community on the data-sharing website that they should download and use the newest version of the dataset for privacy protection.

- **Has an analysis of the potential impact of the dataset and its use on data subjects (e.g., a data protection impact analysis) been conducted?** Yes, see section **??**.

- **Any other comments?** No.

*Preprocessing/cleaning/labeling:*

- **Was any preprocessing/cleaning/labeling of the data done (e.g., discretization or bucketing, tokenization, part-of-speech tagging, SIFT feature extraction, removal of instances, processing of missing values)?** We have checked the self-reporting data of users during data collection. For the information with conflicts, we have filtered out these volunteers in the dataset.

- **Was the "raw" data saved in addition to the preprocessed/cleaned/labeled data (e.g., to support unanticipated future uses)?** No.

- **Is the software used to preprocess/clean/label the instances available?** No, the above data filtering was done manually.

- **Any other comments?** No.

*Uses:*

- **Has the dataset been used for any tasks already?** Not yet.

- **Is there a repository that links to any or all papers or systems that use the dataset?** Not yet, we will supplement this later.

- **What (other) tasks could the dataset be used for?** See Section 4.

- **Is there anything about the composition of the dataset or the way it was collected and preprocessed/cleaned/labeled that might impact future uses?** No.

| Field Name | Type | Description | Example |
|---|---|---|---|
| user_id | numeric | user ID(after hashing), each representing a real user of the platform | 101 |
| pid | numeric | video ID(after hashing), each representing a video collected from the platform | 2023 |
| exposed_time | numeric | Unix timestamp of the interaction | 1663471335 |
| p_date | numeric | date when the interaction happened | 20220917 |
| p_hour | numeric | the hour when the interaction happened | 11 |
| watch_time | numeric | the given user's watching time towards the given video(seconds) | 46 |
| cvm_like | bool | whether the given user gives a like towards the given video | True |
| comment | bool | whether the given user comments for the given video | False |
| follow | bool | whether the given user follows the given video | False |
| collect | bool | whether the given user collects the given video | True |
| forward | bool | whether the given user forwards the given video | False |
| effective_view | bool | whether the watching time surpasses 3 seconds | True |
| hate | bool | whether the given user gives a hate towards the given video | False |

Table 2: Overview of the behavior data.

- **Are there tasks for which the dataset should not be used?** No.

- **Any other comments?** No.

*Distribution:*

- **Will the dataset be distributed to third parties outside of the entity (e.g., company, institution, organization) on behalf of which the dataset was created?** Yes, the dataset is open to the public for relevant research.

- **How will the dataset will be distributed (e.g., tarball on website, API, GitHub)?** The dataset will be distributed with tarball on the provided website.

- **When will the dataset be distributed?** Upon publication.

- **Will the dataset be distributed under a copyright or other intellectual property (IP) license, and/or under applicable terms of use (ToU)?** Similar to ImageNet License.

- **Have any third parties imposed IP-based or other restrictions on the data associated with the instances?** No.

- **Do any export controls or other regulatory restrictions apply to the dataset or to individual instances?** No.

- **Any other comments?** No.

*Maintenance:*

- **Who is supporting/hosting/maintaining the dataset?** Authors of the paper.

- **How can the owner/curator/manager of the dataset be contacted (e.g., email address)?** See the email addresses of the authors of the paper.

- **Is there an erratum?** No.

- **Will the dataset be updated (e.g., to correct labeling errors, add new instances, delete instances)?** It's possible to be updated, e.g., when some users choose to opt out.

- **If the dataset relates to people, are there applicable limits on the retention of the data associated with the instances (e.g., were individuals in question told that their data would be retained for a fixed period of time and then deleted)?** Volunteers can choose to opt out at any time and their information will be erased from the dataset.

- **Will older versions of the dataset continue to be supported/hosted/maintained?** No, upon updating the dataset, it will be noticed on our website.

- **If others want to extend/augment/build on/contribute to the dataset, is there a mechanism for them to do so?** Yes, others who want to extend the dataset can contact the authors and follow the previous data collection process for adding new data to the dataset.

- **Any other comments?** No.

| | Field Name | Type | Description | Example |
|---|---|---|---|---|
| **Video side** | category_level | numeric | the level of category ID(1: primary category; 2: secondary category; 3: Tertiary category) | 3 |
| | category_id | numeric | the tertiary category ID of the video | 1350 |
| | parent_id | numeric | the secondary category ID of the video | 288 |
| | root_id | numeric | the primary category ID of the video | 39 |
| | duration | numeric | duration of the given video(seconds) | 138.566 |
| | author_id | numeric | ID of the video's author(after hashing) | 78 |
| | author_fans_count | numeric | number of fans of the video's author | 138211 |
| | tag_name | text | video's content tag | 'underwater photography' |
| | title | text | title of the video | 'Catch lobsters under the sea' |
| **User side** | gender | text | the user's gender('M': male; 'F': female) | 'M' |
| | age | numeric | the user's age | 34 |
| | mod_price | numeric | the price of user's phones | 1899 |
| | fre_city | text | the city the user locates in | 'Shanghai' |
| | fre_community_type | text | user's residence type | 'country' |
| | fre_city_level | text | user's city level | 'first-tier city' |

Table 3: Overview of the attribute data.

| File | Type | Description | Example |
|---|---|---|---|
| raw_file | folder | all of the raw video files(.mp4) | *2024.mp4*: the raw file of the video with ID "2024" (after hashing) |
| video_feature_total | folder | extracted clip-level visual features of all videos(.npy) | *2024.npy*: an 8*256-dimension tensor, each row is a 256-dimension visual feature for one clip |
| asr_text | folder | text transformed from speech in the video(.txt) | *2024.txt*: the text of the speech recognized from the video "2024.mp4" |

Table 4: Overview of the content data.

| Model | Recall@10 | Recall@20 | Recall@50 | NDCG@10 | NDCG@20 | NDCG@50 |
|---|---|---|---|---|---|---|
| VBPR | 0.0184 | 0.0283 | 0.0519 | 0.0140 | 0.0172 | 0.0237 |
| VBPR (w/o video feature) | 0.0099 | 0.0163 | 0.0315 | 0.0082 | 0.0103 | 0.0148 |
| MMGCN | 0.0105 | 0.0187 | 0.0330 | 0.0088 | 0.0114 | 0.0155 |
| MMGCN (w/o video feature) | 0.0088 | 0.0141 | 0.0254 | 0.0068 | 0.0085 | 0.0117 |
| GRCN | 0.0119 | 0.0224 | 0.0444 | 0.0089 | 0.0125 | 0.0189 |
| GRCN (w/o video feature) | 0.0055 | 0.0112 | 0.0223 | 0.0040 | 0.0058 | 0.0091 |
| BM3 | 0.0238 | 0.0364 | 0.0638 | 0.0178 | 0.0218 | 0.0294 |
| BM3 (w/o video feature) | 0.0231 | 0.0352 | 0.0620 | 0.0171 | 0.0208 | 0.0283 |

Table 5: Performance comparison of 4 typical multimodal recommendation methods with and without video features to validate the practical usage of video features.

## A.3 SUPPLEMENTARY FIGURES AND RESULTS

In this section, we supplement the distribution of the user's implicit and explicit feedback in Figure 8 and Figure 9, and a visual example for the video content information in Figure 11. It can be seen that most volunteers watched within 1,000 seconds, and the videos that have been watched for less than 50 seconds are the most common. Besides, we provide the distribution of wo key video attributes (categories and duration time) in Figure 10a and Figure 10b. In the dataset, most videos are relatively short with less than 30 seconds. Overall, the number of videos decreases with the increasing duration, consistent with the concept of short-video platforms.

## A.4 AUTHOR STATEMENTS

We bear all responsibility in case of violation of rights, etc., and confirmation of the data license.

## A.5 HOSTING, LICENSING, AND MAINTENANCE PLAN

We have established an anonymous website for releasing the dataset which will be open upon publication. We ensure normal access to the data and will provide frequent maintenance and timely updates of any change of the dataset.

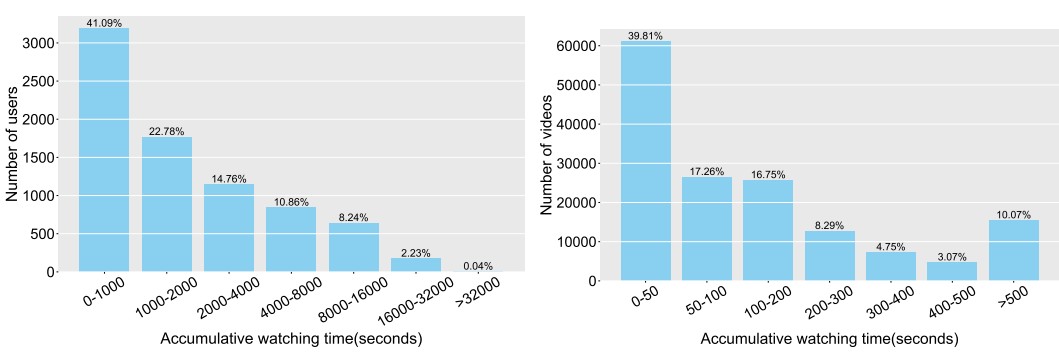

(a) User-side watching time distribution.      (b) Video-side watching time distribution.

Figure 8: Distribution of the implicit feedback(watching time) from (a) user side and (b) video side.

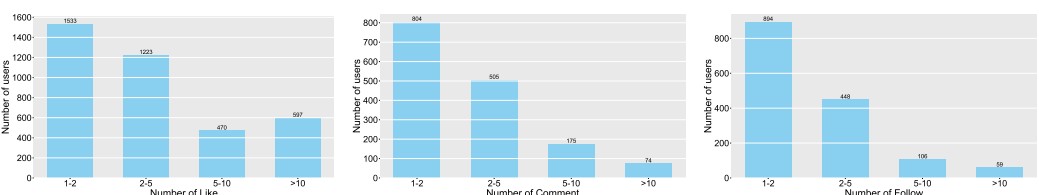

(a) Distribution of like behavior.   (b) Distribution of comment behavior.   (c) Distribution of follow behavior.

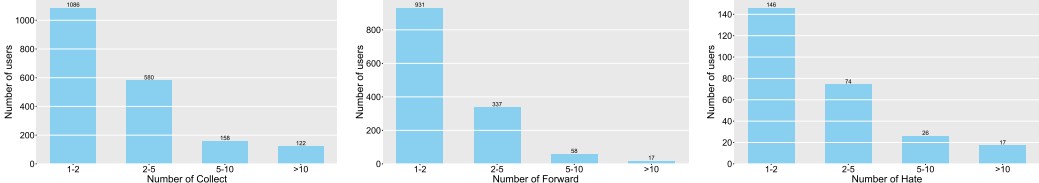

(d) Distribution of collect behavior. (e) Distribution of forward behavior.   (f) Distribution of hate behavior.

Figure 9: Distribution of the explicit feedback including (a) like, (b) comment,(c) follow, (d) collect, (e) forward, (f) hate.

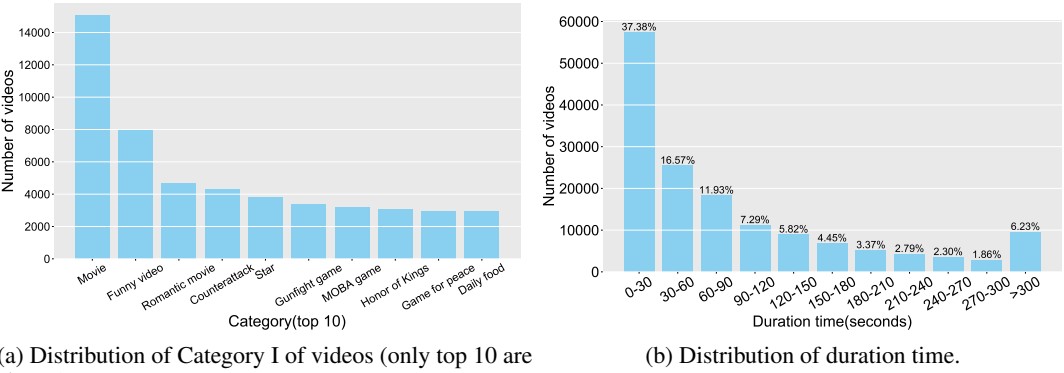

(a) Distribution of Category I of videos (only top 10 are shown)      (b) Distribution of duration time.

Figure 10: Distribution of two key video attributes: (a) Category I and (b) duration time.

## A.6 LIMITATIONS

We discuss some limitations of the dataset as follows. First, the dataset's current scale is smaller compared to some existing recommendation datasets that lack comprehensive attribute and content

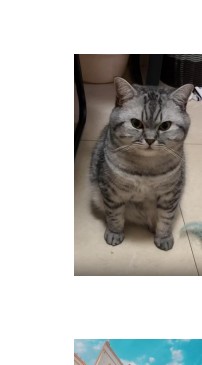

**VideoID:** 16570
**Title:** "Isn't it just not turning on the air conditioner?"
**Tag:** Cat's daily life
**ASR text:** "Ah, isn't that just not turning on the air conditioner for you? You take off your clothes and that's it, right? You're in a heat wave of 40 degrees, and you're wearing a leather jacket. Can you not get hot? "
**Category I:** wildlife
**Category II:** pets
**Category III:** cat
**Duration:** 8s

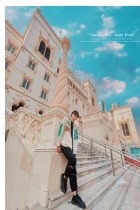

**VideoID:** 110124
**Title:** "How to shoot a big movie with European architecture"
**Tag:** Creative inspiration, Celebrity photo poses
**ASR text:** "Weekend travel, saw such beautiful European-style buildings, don't let your friends take such ridiculous poses again. The angle is really unbearable. Two small skills teach you to take big shots. First, we open the camera and choose 0.5 times super wide angle, we turn the phone upside down to shoot, use the three-point line composition…."
**Category I:** None
**Category II:** post-processing
**Category III:** teaching photography
**Duration:** 35.566s

Figure 11: Visual examples of video content in our dataset.

data. This limitation arises due to the challenges of collecting data from volunteers. Second, the dataset primarily focuses on users from a single country. Expanding its coverage to include users from multiple countries would further enhance its generalizability. Third, the amount of explicit user feedback is limited, as the data was collected over a relatively short period. In the future, we will extend the scale and user coverage of our dataset and update it timely.

