# OpenReview forum: "A Large-scale Dataset with Behavior, Attributes, and Content of Mobile Short-video Platform"
_ICLR.cc/2025/Conference — ICLR 2025 Conference Withdrawn Submission_

### Official Review · Reviewer_WAdp · 2024-10-29

**Soundness:** 3
**Presentation:** 3
**Contribution:** 1
**Rating:** 1
**Confidence:** 5

**Summary:**

This paper introduces a rich dataset collected from a real mobile short-video platform, filling important gaps in data on user behavior, demographics, and video content. With information from 10,000 users and over 153,000 videos, it’s a valuable resource for researchers in social science and AI.

- The dataset captures over a million user interactions, covering both passive behaviors (like viewing time) and active actions (likes, comments, follows, shares, saves, and dislikes), which enables a deeper look into user interests and behaviors.
- It includes detailed demographic information, such as age, gender, city, city size, community type, and device price, allowing researchers to explore how different user groups engage with the platform.
- For video content, the dataset includes raw video files, visual features, and transcripts generated through speech-to-text, organized into primary, secondary, and tertiary categories for easy content analysis.
- Extensive validation ensures that the dataset accurately represents various user behaviors and demographics. Video content quality is confirmed through visual clustering techniques, which show clear distinctions among video types.
- This dataset stands out for its scale and depth, surpassing similar datasets like Kuaishou, REASONER, and MicroVideo-1.7M, and is ideal for studies on user modeling, recommendation systems, and topics like filter bubbles and user addiction.
- Designed to support diverse research areas, the dataset can be used to improve personalized recommendations, study fairness in algorithms, understand user engagement, and examine how recommendation systems influence information diversity and polarization.
- It’s publicly available and ethically compliant, with user and video identifiers anonymized, and all sensitive data handled carefully to protect privacy.

**Strengths:**

- The dataset integrates detailed user behavior logs, diverse user and video attributes, and raw video content, offering a comprehensive foundation for research across various fields. This thorough approach addresses gaps found in other datasets, making it a valuable resource for studies in computational social science, artificial intelligence, and algorithmic fairness.

- With the inclusion of raw video files alongside processed visual features and automatic speech recognition (ASR) text, the dataset enables deep video understanding and multimodal research, expanding analytical capabilities beyond what many other datasets support.

- Containing over one million interactions from 10,000 users across 153,000 videos, the dataset is robust, dependable, and well-suited for various analytical tasks. It provides in-depth demographic, geographical, and device-related data that supports research in recommendation systems, social science, and algorithmic fairness, while minimizing statistical biases.

- The dataset is structured with clarity, including detailed tables and figures to enhance accessibility and usability. Ethical considerations, such as informed consent, anonymized data, and opt-out options, uphold high privacy standards, making the dataset a trustworthy resource for the research community.

- Designed for versatility, this dataset supports a wide range of studies, including user modeling, AI fairness, filter bubbles, polarization, and user addiction, making it a valuable asset across multiple research domains.

**Weaknesses:**

- The paper does not compare model performance trained on this new dataset with previous benchmarks in the literature, making it challenging to assess the dataset’s practical utility. This absence of baseline comparisons limits our ability to measure performance gaps and evaluate how well (or poorly) this dataset performs relative to established standards. For me, this is the most critical aspect, as such a comparison is essential to determine the dataset's true impact and value in advancing the field.

- The selection criteria for users and videos are not explicitly detailed, raising concerns about the dataset's representativeness and potential sampling biases. The paper does not  also sufficiently analyze or acknowledge potential biases, such as volunteer self-selection bias, geographic concentration, or device type distribution, which could affect the dataset's generalizability.

- Apart from qualitative t-SNE visualizations, the study lacks a quantitative evaluation, such as clustering metrics or downstream task performance, to effectively validate the quality of the extracted video features.

- The review of existing datasets and related work is not exhaustive, potentially overlooking relevant studies and resources that could contextualize the dataset's contributions.

**Questions:**

- Could you provide baseline experiments demonstrating the dataset's applicability and effectiveness for common tasks such as recommendation, user modeling, or video classification?

- Expand the literature review to encompass the latest publicly available short-video datasets. Provide a comparative table highlighting key features, such as the number of users, videos, interaction types, and content richness, to clearly demonstrate how your dataset stands out.

- Please adjust Figure 4, as the text appears too close together. Changing the format would improve its readability.

- It would be helpful to enlarge the legends in Figures 3, 4, and 5, as they are currently quite small. Also, please consider using a color palette that is accessible to colorblind viewers, including in Figure 6.

- Include case studies or illustrative experiments that showcase the dataset's application in areas such as user modeling, fairness analysis, or studying filter bubbles. Presenting preliminary results or hypothetical scenarios can help researchers envision practical uses and inspire innovative applications.

- Describe the handling of multilingual content within the dataset. Specify the languages supported by the ASR system and outline any post-processing steps taken to enhance transcription accuracy. Provide statistics on the accuracy rates or error rates of the ASR results, if available.

- How do the potential biases affect the dataset's generalizability, and what steps have been taken to mitigate them?

- Add a dedicated section to discuss the dataset's limitations, covering potential biases, data sparsity in specific interaction types, and gaps in demographic coverage. Transparently addressing these issues helps researchers to consider them in their analyses and supports trust in the dataset's integrity.

---

> ### Author Response · Authors · 2024-11-25
> **Response to Reviewer WAdp (part 1)**
>
> **Q1: The paper does not compare model performance trained on this new dataset with previous benchmarks in the literature, making it challenging to assess the dataset’s practical utility. This absence of baseline comparisons limits our ability to measure performance gaps and evaluate how well (or poorly) this dataset performs relative to established standards. For me, this is the most critical aspect, as such a comparison is essential to determine the dataset's true impact and value in advancing the field.**
>
> **Response:** We have supplemented experimental results of 8 recommendation algorithms (including both general and multimodal recommendation methods) to validate the practical usage of our dataset on user modeling. The results are as follows:
>
> |  Model   | Recall@10 | Recall@20 | Recall@50 | NDCG@10 | NDCG@20 | NDCG@50 |
> | :------: | :-------: | :-------: | :-------: | :-----: | :-----: | :-----: |
> |   BPR    |  0.0113   |  0.0218   |  0.0453   | 0.0078  | 0.0114  | 0.0179  |
> | LightGCN |  0.0223   |  0.0358   |  0.0629   | 0.0169  | 0.0211  | 0.0286  |
> | LayerGCN |  0.0208   |  0.0336   |  0.0559   | 0.0160  | 0.0198  | 0.0260  |
> |   VBPR   |  0.0184   |  0.0283   |  0.0519   | 0.0140  | 0.0172  | 0.0237  |
> |  MMGCN   |  0.0105   |  0.0187   |  0.0330   | 0.0088  | 0.0114  | 0.0155  |
> |   GRCN   |  0.0119   |  0.0224   |  0.0444   | 0.0089  | 0.0125  | 0.0189  |
> |   BM3    |  0.0238   |  0.0364   |  0.0638   | 0.0178  | 0.0218  | 0.0294  |
> |  LGMRec  |  0.0159   |  0.0257   |  0.0449   | 0.0131  | 0.0162  | 0.0217  |
>
> **Q2: The selection criteria for users and videos are not explicitly detailed, raising concerns about the dataset's representativeness and potential sampling biases. The paper does not also sufficiently analyze or acknowledge potential biases, such as volunteer self-selection bias, geographic concentration, or device type distribution, which could affect the dataset's generalizability.**
>
> **Response:**
>
> We have conducted the validation of the bias issue on two aspects.
>
> - **Validation of the macroscopic data distribution.** We have checked the consistency between the user-side data of our dataset and the public information released by the short-video platform. According to the official report, there were users with 44% female and 56% male. In our dataset, there are 43% female and 57% male, which is consistent with the real user distribution on the platform without obvious bias. This provides evidence for the reliability of the overall collected data, and thus other user-side attributes can also be ensured unbiased.
> - **Validation of the microscopic specific data.** We have addressed the bias issue of specific user-side data on two aspects. **On the one hand**, we have introduced internal consistency checking in the survey to estimate how consistent the results are for different questions about the same fact. If volunteers were to randomly report their information, the probability of passing the internal consistency check is extremely low. While actually we find no obvious failure of the internal consistency checking during the data collection, meaning that the self-reporting data is actually accurate. **On the other hand**, we have compared the self-reporting data with the public user profile on the platform uploaded by themselves (*e.g.,* gender, age) and information recorded by the proxy agent we use (*e.g.,* device). We found that there's nearly no conflicting information, meaning that these data is originally unbiased. These treatments have already shown the overall reliability of users' self-reporting data. Besides, we have further filtered the very few unreliable data to keep the dataset unbiased.
>
> Moreover, considering that the volunteer group is large enough and covers a wide range of user attributes, the experimental group can be already representative. Therefore, the representativeness of the dataset is not a big issue.

---

> ### Author Response · Authors · 2024-11-25
> **Response to Reviewer WAdp (part 2)**
>
> **Q3: Apart from qualitative t-SNE visualizations, the study lacks a quantitative evaluation, such as clustering metrics or downstream task performance, to effectively validate the quality of the extracted video features.**
>
> **Response:** We have tested 4 typical multimodal recommendation methods with and without video features to validate the practical usage of these features. From the results, it can be seen that there's common performance drop without video features of all methods, validating the quality of extracted video features.
>
> |           Model           | Recall@10 | Recall@20 | Recall@50 | NDCG@10 | NDCG@20 | NDCG@50 |
> | :-----------------------: | :-------: | :-------: | :-------: | :-----: | :-----: | :-----: |
> |           VBPR            |  0.0184   |  0.0283   |  0.0519   | 0.0140  | 0.0172  | 0.0237  |
> | VBPR (w/o video feature)  |  0.0099   |  0.0163   |  0.0315   | 0.0082  | 0.0103  | 0.0148  |
> |           MMGCN           |  0.0105   |  0.0187   |  0.0330   | 0.0088  | 0.0114  | 0.0155  |
> | MMGCN (w/o video feature) |  0.0088   |  0.0141   |  0.0254   | 0.0068  | 0.0085  | 0.0117  |
> |           GRCN            |  0.0119   |  0.0224   |  0.0444   | 0.0089  | 0.0125  | 0.0189  |
> | GRCN (w/o video feature)  |  0.0055   |  0.0112   |  0.0223   | 0.0040  | 0.0058  | 0.0091  |
> |            BM3            |  0.0238   |  0.0364   |  0.0638   | 0.0178  | 0.0218  | 0.0294  |
> |  BM3 (w/o video feature)  |  0.0231   |  0.0352   |  0.0620   | 0.0171  | 0.0208  | 0.0283  |
>
> **Q4: The review of existing datasets and related work is not exhaustive, potentially overlooking relevant studies and resources that could contextualize the dataset's contributions.**
>
> **Response:**
>
> We have provided a comparison table about relevant short-video datastes as follows. In summary, a core limitation of existing datasets is the limited content richness. Microvideo-1.7M [1] provides only video category information, while datasets like KuaiRec [2], REASONER [3] and Tenrec [4] include only video text data or pre-extracted visual features from video thumbnails. Although the more recent Microlens [5] dataset includes raw videos, it is constrained by a limited number of videos. More critically, it lacks rich user-side information, such as demographic and geographic attributes, offering only user ID data. By comparison, our dataset is more comprehensive and balanced on content richness and data scale.
>
> | Dataset                           | User Num  | Video Num | Interaction Num | Interaction types                                            | Content richness                                             |
> | --------------------------------- | --------- | --------- | --------------- | ------------------------------------------------------------ | ------------------------------------------------------------ |
> | MicroVideo-1.7M                   | 10,986    | 1,704,880 | 12,737,619      | click, observe but not click                                 | one-hot video category                                       |
> | KuaiRec                           | 1,411     | 3,327     | 4,676,570       | watching time, watch_ratio                                   | video_type, upload information, video categories, video duration, tag, caption, music_id, cover text |
> | REASONER                          | 2,997     | 4,672     | 58,497          | like, rating, review, tag selection, watch again             | title, tag, category, duration, introduction                 |
> | Tenrec (QK-Video)                 | 5,022,750 | 3,753,436 | 142,321,193     | click, like, share, follow                                   | video type feature                                           |
> | Microlens (released 100K version) | 100,000   | 19,738    | 719,405         | click, like                                                    | raw video, title, audio, cover images                        |
> | Ours                              | 10,000    | 153,561   | 1,019,568       | like, follow, forward, collect, comment, hate, watching time, effective_view | video categories, duration, tag, title, raw videos, extracted video feature, ASR text |
>
> [1] Chen, Xusong, et al. "Temporal hierarchical attention at category-and item-level for micro-video click-through prediction." *Proceedings of the 26th ACM international conference on Multimedia*. 2018.
>
> [2] Gao, Chongming, et al. "KuaiRec: A fully-observed dataset and insights for evaluating recommender systems." *Proceedings of the 31st ACM International Conference on Information & Knowledge Management*. 2022.
>
> [3] Chen, Xu, et al. "REASONER: an explainable recommendation dataset with comprehensive labeling ground truths." NIPS 2024.
>
> [4] Yuan, Guanghu, et al. "Tenrec: A large-scale multipurpose benchmark dataset for recommender systems." NIPS 2022.
>
> [5] Ni, Yongxin, et al. "A content-driven micro-video recommendation dataset at scale." *arXiv preprint arXiv:2309.15379* (2023).

---

> ### Author Response · Authors · 2024-11-25
> **Response to Reviewer WAdp (part 3)**
>
> **Q5: Could you provide baseline experiments demonstrating the dataset's applicability and effectiveness for common tasks such as recommendation, user modeling, or video classification?**
>
> **Response:**  We have supplemented experimental results of 8 recommendation algorithms, please see the response to Q1.
>
> **Q6: Expand the literature review to encompass the latest publicly available short-video datasets. Provide a comparative table highlighting key features, such as the number of users, videos, interaction types, and content richness, to clearly demonstrate how your dataset stands out.**
>
> **Response:** Please see the response to Q4.
>
> **Q7: Please adjust Figure 4, as the text appears too close together. Changing the format would improve its readability.**
>
> **Response:** Thanks for the suggestion and we have corrected that in the revised paper.
>
> **Q8: It would be helpful to enlarge the legends in Figures 3, 4, and 5, as they are currently quite small. Also, please consider using a color palette that is accessible to colorblind viewers, including in Figure 6.**
>
> **Response:** Thanks for the suggestion and we have corrected that in the revised paper.
>
> **Q9: Include case studies or illustrative experiments that showcase the dataset's application in areas such as user modeling, fairness analysis, or studying filter bubbles. Presenting preliminary results or hypothetical scenarios can help researchers envision practical uses and inspire innovative applications.**
>
> **Response:** Thanks for the suggestion. We have supplemented experimental results of 8 recommendation algorithms to validate the practical usage of our dataset on user modeling. The results are shown in the response to Q1.
>
> As for social science and human behavior understanding research, we provide some preliminary ideas for studying filter bubble phenomenon with our dataset as an example. The filter bubble refers to the situation where recommendation algorithms overly tailor content to a user's preference, resulting in limited content exposure for the user on the platform. To quantify content coverage, our dataset includes a hierarchical content category system. Coverage at a category level c is defined as
>
> $$
> \frac{N_{seen}(u,c)}{N_{all}(c)}
> $$
> where N_seen represents the number of level-c categories a user has interacted with over a period, and N_all is the total number of level-c categories available. A user's filter bubble extent can then be determined by comparing this metric against a threshold, such as the average across all users. Subsequent analyses could investigate how user behaviors, such as viewing time and content preferences, influence the severity of filter bubbles across different category levels. Additionally, tracking the temporal evolution of filter bubbles at each level could provide insights into their dynamics and persistence over time.
>
> **Q10: Describe the handling of multilingual content within the dataset. Specify the languages supported by the ASR system and outline any post-processing steps taken to enhance transcription accuracy. Provide statistics on the accuracy rates or error rates of the ASR results, if available.**
>
> **Response:** We utilized an advanced model SenceVoice-Small for speech recognition and LLaMA-3-8B for translation, obtaining bilingual (Chinese and English) ASR texts. We further conduct manual examination and refinement of the obtained text to ensure the data quality.
>
> **Q11: How do the potential biases affect the dataset's generalizability, and what steps have been taken to mitigate them?**
>
> **Response:** Please refer to the response to Q2.
>
> **Q12: Add a dedicated section to discuss the dataset's limitations, covering potential biases, data sparsity in specific interaction types, and gaps in demographic coverage. Transparently addressing these issues helps researchers to consider them in their analyses and supports trust in the dataset's integrity.**
>
> **Response:** We discuss some limitations of the dataset as follows, which is also supplemented in Section A.6 of the Appendix. First, the dataset's current scale is smaller compared to some existing recommendation datasets that lack comprehensive attribute and content data. This limitation arises due to the challenges of collecting data from volunteers. Second, the dataset primarily focuses on users from a single country. Expanding its coverage to include users from multiple countries would further enhance its generalizability. Third, the amount of explicit user feedback is limited, as the data was collected over a relatively short period. In the future, we will extend the scale and user coverage of our dataset and update it timely.

---

> > ### Comment · Reviewer_WAdp · 2024-11-26
> > **Reply to the Authors**
> >
> > **Q1** The fact that results are still being obtained on different models long after the deadline makes me think that the paper is not yet ready. **I also agree with what reviewer HpcX mentioned:** although new experimental results have been included, the relevance of these results to the dataset and the validation of its value remain very limited. Additionally, the new results seem somewhat low. It would be helpful to include, since this is a Recall metric, what the recall of random choice would be, to provide a baseline measure and ensure that the model is not simply producing random outputs.
> >
> > **Q2** It's a valid argument. It helped to solve my concern.
> >
> > **Q3** Random Choice baseline is missing and the oracle (upper bond) too. These two are good clues to determine the limitations of the dataset.
> >
> > **Q4** This study is quite interesting and addresses aspects that were missing in the original paper. However, I believe the paper could benefit from further refinement before it is fully ready. These valuable new insights could be a great addition to a future submission, allowing the work to be even more comprehensive and impactful.
> >
> > **Q5, Q6, Q7, Q8**  The authors addressed my concerns accordingly.
> >
> > **Q9**  I consider that this point should be discussed in the paper, and it would be beneficial to include it in the next revision.
> >
> > **Q10** These details should be added step by step, at least in the supplementary material, to ensure that the experiments can be reproduced.
> >
> > **Q11, Q12** The authors addressed my concerns accordingly.
> >
> > The authors have done a good job addressing many of my questions, and I appreciate their efforts. However, the addition of quantitative analyses after the deadline raises some concerns about the readiness of the paper. I recommend that the reviewers consider all the discussions here and incorporate them into the paper, as this would significantly strengthen it for a future revision. While I value the progress made, I still agree with reviewer HpcX’s assessment. With the suggested changes, the paper has the potential to improve substantially in the next revision, but at this stage, I feel it is not yet ready for ICLR.

---

### Official Review · Reviewer_HpcX · 2024-10-31

**Soundness:** 2
**Presentation:** 2
**Contribution:** 2
**Rating:** 5
**Confidence:** 3

**Summary:**

This paper introduces a large-scale dataset derived from a mobile short-video platform. Compared to previous datasets, it expands the scope of user behavior data (including both explicit and implicit feedback), user attributes (covering demographics, geographic, and device-related information), and video content (including raw video files, visual features, and ASR transcripts). The paper provides a technical validation of the dataset, confirming the comprehensive nature of its behavioral data, broad attribute coverage, and representational strength of its content features.

**Strengths:**

1. The dataset includes raw videos, which is valuable as many video datasets lack this component today.
2. The dataset captures user actions and attributes from multiple perspectives, offering new insights into the relationship between user behavior and video content.

**Weaknesses:**

1. The link to the dataset seems to be broken, possibly due to a network issue on my end. Could you provide guidance on accessing it? I may consider raising the score if access is successful.
2. From a data perspective:Since the data originates from a single social media platform, user behavior is influenced by the platform’s
    * recommendation algorithm. This introduces significant bias due to these recommendation effects.
    * The visual features were extracted using a 2016 image model, while more advanced models are now available that can extract features from video encodings. Thus, the extracted visual features may be insufficient.
3. From a validation perspective:
    * The conclusions presented in Section 3.1.2 lack depth and seem self-evident.
    * The paper lacks proof-of-concept experiments to demonstrate the dataset’s value in the proposed research areas, which somewhat diminishes the paper’s impact.
    * Most validations rely on statistical analyses, rather than truly testing the representational ability of the video data. Statistical data alone is insufficient to prove the dataset’s research value.
4. Overall, while the dataset provides fresh data and unique insights into user actions, the paper lacks experimental results demonstrating the dataset’s applicability across the various domains it proposes.

**Questions:**

see weakness

---

> ### Author Response · Authors · 2024-11-25
> **Response to Reviewer HpcX (part 1)**
>
> **Q1: The link to the dataset seems to be broken, possibly due to a network issue on my end. Could you provide guidance on accessing it? I may consider raising the score if access is successful.**
>
> **Response:** Sorry for the problem. We now provide another link for your review: https://www.dropbox.com/scl/fo/5z7pwp6xjkrr1926vreu6/AFYter5C6BDTOCpxkxF0k9Y?rlkey=p28j6u1fl1ubb7bufiq16onbl&st=7aktff85&dl=0, in which we sample 10 video files with corresponding title text, ASR text, visual feature data and the full interaction data. Details about the shown data are as follows:
>
> - **raw\_file:** the folder containing original video files with the format "video_id.mp4".
>
> - **video\_feature\_total:** the folder containing all extracted visual features with the format "video_id.npy".
>
> - **interaciton.csv:** the file containing all behavior data and attribute data.
>
> - **asr_zn:** ASR text in Chinese of each video with the format "video_id.txt".
>
> - **asr_en:** ASR text in English of each video with the format "video_id.txt".
>
> - **title_en:** video title text in English with the format "video_id.txt".
>
> - **categories_cn_en.csv:** the name of categories in English and Chinese.
>
> **Q2: From a data perspective: Since the data originates from a single social media platform, user behavior is influenced by the platform’s**
>
> - **recommendation algorithm. This introduces significant bias due to these recommendation effects.**
> - **The visual features were extracted using a 2016 image model, while more advanced models are now available that can extract features from video encodings. Thus, the extracted visual features may be insufficient.**
>
> **Response:**
>
> **About the bias:** We agree that exposure bias is an inherent challenge in most offline recommendation system datasets, including our dataset. This bias arises naturally from real-world application scenarios where users are exposed to limited content. As such, all datasets with actual user interactions are likely to exhibit some degree of exposure bias, it mirrors the conditions under which recommendation systems operate in practice. Therefore, addressing exposure bias is not merely a matter of dataset but requires more effort from advanced recommendation algorithms (e.g., some causal inference methods).
>
> **About the visual features:** We have used a more advanced model ViT-base for visual feature extraction, and the files are updated.
>
> **Q3: From a validation perspective:**
>
> - **The conclusions presented in Section 3.1.2 lack depth and seem self-evident.**
> - **The paper lacks proof-of-concept experiments to demonstrate the dataset’s value in the proposed research areas, which somewhat diminishes the paper’s impact.**
> - **Most validations rely on statistical analyses, rather than truly testing the representational ability of the video data. Statistical data alone is insufficient to prove the dataset’s research value.**
>
> **Response:**
>
> **About analysis in Section 3.1.2:** We have added more detailed analysis about relations between different user behaviors, please refer to Section 3.1.2 in the updated paper.
>
> **About proof-of-concept experiments:** We have added experiments of recommendation algorithms and preliminary ideas for studying filter bubble with our dataset, please refer to the response to Q4 for more details.
>
> **About validation of the representation ability of the video data:** We have tested 4 typical multimodal recommendation methods with and without video features to validate the practical usage of these features. From the results, it can be seen that there's common performance drop without video features of all methods, demonstrating that the value of extracted video features.
>
> |           Model           | Recall@10 | Recall@20 | Recall@50 | NDCG@10 | NDCG@20 | NDCG@50 |
> | :-----------------------: | :-------: | :-------: | :-------: | :-----: | :-----: | :-----: |
> |           VBPR            |  0.0184   |  0.0283   |  0.0519   | 0.0140  | 0.0172  | 0.0237  |
> | VBPR (w/o video feature)  |  0.0099   |  0.0163   |  0.0315   | 0.0082  | 0.0103  | 0.0148  |
> |           MMGCN           |  0.0105   |  0.0187   |  0.0330   | 0.0088  | 0.0114  | 0.0155  |
> | MMGCN (w/o video feature) |  0.0088   |  0.0141   |  0.0254   | 0.0068  | 0.0085  | 0.0117  |
> |           GRCN            |  0.0119   |  0.0224   |  0.0444   | 0.0089  | 0.0125  | 0.0189  |
> | GRCN (w/o video feature)  |  0.0055   |  0.0112   |  0.0223   | 0.0040  | 0.0058  | 0.0091  |
> |            BM3            |  0.0238   |  0.0364   |  0.0638   | 0.0178  | 0.0218  | 0.0294  |
> |  BM3 (w/o video feature)  |  0.0231   |  0.0352   |  0.0620   | 0.0171  | 0.0208  | 0.0283  |

---

> ### Author Response · Authors · 2024-11-25
> **Response to Reviewer HpcX (part 2)**
>
> **Q4:** Overall, while the dataset provides fresh data and unique insights into user actions, the paper lacks experimental results demonstrating the dataset’s applicability across the various domains it proposes.
>
> **Response:** We have supplemented experimental results of 8 recommendation algorithms (including both general and multimodal recommendation methods) to validate the practical usage of our dataset on user modeling. The results are as follows:
>
> |  Model   | Recall@10 | Recall@20 | Recall@50 | NDCG@10 | NDCG@20 | NDCG@50 |
> | :------: | :-------: | :-------: | :-------: | :-----: | :-----: | :-----: |
> |   BPR    |  0.0113   |  0.0218   |  0.0453   | 0.0078  | 0.0114  | 0.0179  |
> | LightGCN |  0.0223   |  0.0358   |  0.0629   | 0.0169  | 0.0211  | 0.0286  |
> | LayerGCN |  0.0208   |  0.0336   |  0.0559   | 0.0160  | 0.0198  | 0.0260  |
> |   VBPR   |  0.0184   |  0.0283   |  0.0519   | 0.0140  | 0.0172  | 0.0237  |
> |  MMGCN   |  0.0105   |  0.0187   |  0.0330   | 0.0088  | 0.0114  | 0.0155  |
> |   GRCN   |  0.0119   |  0.0224   |  0.0444   | 0.0089  | 0.0125  | 0.0189  |
> |   BM3    |  0.0238   |  0.0364   |  0.0638   | 0.0178  | 0.0218  | 0.0294  |
> |  LGMRec  |  0.0159   |  0.0257   |  0.0449   | 0.0131  | 0.0162  | 0.0217  |
>
> As for social science and human behavior understanding research, we provide some preliminary ideas for studying filter bubble phenomenon with our dataset as an example. The filter bubble refers to the situation where recommendation algorithms overly tailor content to a user's preference, resulting in limited content exposure for the user on the platform. To quantify content coverage, our dataset includes a hierarchical content category system. Coverage at a category level c is defined as
>
> $$
> \frac{N_{seen}(u,c)}{N_{all}(c)}
> $$
> where N_seen represents the number of level-c categories a user has interacted with over a period, and N_all is the total number of level-c categories available. A user's filter bubble extent can then be determined by comparing this metric against a threshold, such as the average across all users. Subsequent analyses could investigate how user behaviors, such as viewing time and content preferences, influence the severity of filter bubbles across different category levels. Additionally, tracking the temporal evolution of filter bubbles at each level could provide insights into their dynamics and persistence over time.

---

> ### Comment · Reviewer_HpcX · 2024-11-25
> **Reply to Author**
>
> Thank you for your effort and reply!
>
> ---
>
> **For Q1:** The update of the dataset link has resolved my concern.
>
> ---
>
> **For Q2:** My concern has not been addressed. Since the goal is to study user behavior while watching videos on social media, it is crucial to consider that most users typically use multiple platforms. Sampling data from multiple platforms would effectively mitigate such biases. I believe this type of bias is critical when constructing a user behavior dataset. Research based on biased data can lead to conclusions that are limited by the constraints of a single platform.
>
> ---
>
> **For Q3:**
>
> 1. In the updated version of the paper, I did not find Section 3.1.2. Additionally, if modifications were made, it would be better to highlight them in a different color for clarity.
> 2. Response in Q4 rebuttal.
> 3. I find the comparison insufficient. I believe the dataset’s proposed video representations should be compared with other video representations to demonstrate their role and distinctions better.
>
> ---
>
> **For Q4:**
>
> 1. The result of the recommendation system is a good example to help me understand the value of your data.
> 2. Since the formula you proposed makes sense, why not conduct some experiments based on your datasets and analysis instead of merely presenting assumptions?
> ---
>
> Although additional experimental results have been included, the relevance of these results to the dataset and the validation of its value remains very limited.

---

### Official Review · Reviewer_wKmC · 2024-11-02

**Soundness:** 3
**Presentation:** 3
**Contribution:** 3
**Rating:** 6
**Confidence:** 4

**Summary:**

The paper presents a new collected dataset for user's interaction with the social media platform, TikTok. The dataset contains data about the videos (raw pixels), video's metadata, user's metadata, and multiple user's explicit and implicit interactions with each of the videos. The data is mostly (if not entirely) covering Chinese users from multiple demographics (data which is also provided).
This dataset is the first one to release all these information along with raw video (audio, frames, ASR) data, which can be useful for multiple applications. The value of the data lies on the fact that will be made publicly available, providing a step forward democratizing the study of recommendations algorithms and their implications.

**Strengths:**

The paper's main contribution is the data. The richness of the data collection and the fact that the authors have consent to use and distribute this data to the research community is a very valuable contribution for future studies on recommendation algorithms, biases in them and the potential societal impact that these might have.

The authors provide several initial insights on the data that give an overview of the potential of it.

The collection pipeline is also very valuable as it can serve for future datasets to follow the same protocol to enrich the data further to more diverse demographics.

**Weaknesses:**

The paper has two main weaknesses that I would like the authors to discuss carefully:
1. Every dataset has biases. However, this one in particular has a very critical one which is the population that it focused on. In Figure 5d, it's clear that the users that appear in the dataset are mostly from China, which makes the dataset tailored to a single demographic population. Although, the data is still valuable, conclusions that will be made in the future using this data can be only applied to the Chinese population. I would like to know what to the authors and other reviewers think about this critical point.

2. Most of the data provided by the data is simply collected by how the users interact with the platform. However, there are a couple of design choices that were unclear to me:

     - There is no explanation on how the video categories I,II, and III were chosen. The authors mention that they are hierarchical, which sounds like a good idea. However, the is no explanation on how the hierarchies were chosen and how were the classes picked from the tags, titles, and content of the videos. How was this discretization of classes done for I? How was then the hierarchical structure form to have II and III?

    - It was not clear to me what is the "effective view" label, what does it try to convey and why is the threshold of 3 seconds picked?



Others:
although not a weakness I have a few recommendations to improve the presentation of the paper.

1. Whenever not sure about the gender of the referring person, it's better to use THEY as pronoun instead of saying he/she all the time. It also easier to read this way.
2. Figure 4 x axis should number of like**s** in plural.
3. Figure 4 caption can be further improved by stating that the data presented is per user not per video, it can be confusing if one reads the figure before reaching the paragraph in which it is mentioned.
4. Figure can be moves so they are closer to the place they are mentioned. Right now Figures 3 and 4, are like 2 to 3 pages away from the paragraph they are mentioned in. It would be better if authors manage to arrange them in a way that they are closer to the sections they belong to.

**Questions:**

1. What are the implications on the fact that most of the users from the dataset come from a single country? What are the benefits from it? What are the downsides of having so little diversity in terms of country of origin? Could the conclusions made with this data be extrapolated to other places? If not, what could be the solution for this?

2. What was the criteria to pick the classes for Category I?

3. What was then the Criteria to make the hierarchies from I to II and III?

**Details Of Ethics Concerns:**

Although the authors have been very transparent about their data collection and statistics. I still think that a closer look to the data release and the biases on the dataset should be checked. I am no expert in these topics and from my perspective looks like the authors have been transparent and have everything in order for the release of the data. However, I still think that a closer look into it would be beneficial given that the dataset might contain sensitive information about the users involved and their preferences and interactions with the platform.

---

> ### Author Response · Authors · 2024-11-25
> **Response to Reviewer wKmC (part 1)**
>
> **Q1: Every dataset has biases. However, this one in particular has a very critical one which is the population that it focused on. In Figure 5d, it's clear that the users that appear in the dataset are mostly from China, which makes the dataset tailored to a single demographic population. Although, the data is still valuable, conclusions that will be made in the future using this data can be only applied to the Chinese population. I would like to know what to the authors and other reviewers think about this critical point.**
>
> **Response:** Collecting a comprehensive dataset of short-video platforms with behavior, attributes and content data from volunteers is actually very challenging, even within a single country. Although the population are most Chinese, our dataset has covered diverse user groups in terms of demographic and geographic attributes, as detailed in Section 3.2 and Figure 5 in the main text. Therefore, the data is already rich and diverse, and the generalization ability can be ensured to some extent. Moreover, it is important to note that many existing datasets, such as Kuairec [1], REASONER [2], Tenrec [3] and Microlens [4], similarly focus on a single platform and primarily cover users from one country. This is often due to the practical challenges of cross-regional data collection. We are actively working to expand our data collection efforts to include users from other countries to further enhance the population coverage.
>
> [1] Gao, et al. "KuaiRec: A fully-observed dataset and insights for evaluating recommender systems", CIKM 2022.
>
> [2] Xu, et al. "REASONER: An Explainable Recommendation Dataset with Multi-aspect Real User Labeled Ground Truths Towards more Measurable Explainable Recommendation", arXiv 2023.
>
> [3] Yuan, et al. "Tenrec: A large-scale multipurpose benchmark dataset for recommender systems." NeurIPS 2022.
>
> [4] Ni, Yongxin, et al. "A content-driven micro-video recommendation dataset at scale." arXiv 2023.
>
> **Q2: Most of the data provided by the data is simply collected by how the users interact with the platform. However, there are a couple of design choices that were unclear to me:**
>
> - **There is no explanation on how the video categories I,II, and III were chosen. The authors mention that they are hierarchical, which sounds like a good idea. However, the is no explanation on how the hierarchies were chosen and how were the classes picked from the tags, titles, and content of the videos. How was this discretization of classes done for I? How was then the hierarchical structure form to have II and III?**
> - **It was not clear to me what is the "effective view" label, what does it try to convey and why is the threshold of 3 seconds picked?**
>
> **Response:**
>
> The process for defining the video categories (I, II, and III) was carried out iteratively as follows:
>
> **Category I Formation**: Videos were first clustered based on similarities in their title and tag texts. Each cluster was manually reviewed to define the overarching theme it represented, leveraging human expertise to finalize the primary categories. To ensure accuracy, every video was manually checked to confirm that its assigned category matched its actual content, with misclassified videos corrected.
>
> **Category II Formation**: For each cluster within Category I, the process above was repeated. Videos in each primary category were further clustered, manually reviewed, and refined to define the secondary subcategories.
>
> **Category III Formation**: The same procedure was applied to each secondary category, further refining the classification to form tertiary categories.
>
> **About "effective view":** Users passively engage with recommended videos on the platform, and they may quickly skip over certain videos, especially those they find uninteresting. This label provides an implicit metric to measure users' genuine attitudes toward the content they watch. As for the 3 seconds threshold, it's based on our observation and experience. Considering that the dataset includes original watching time, this threshold can be adjusted according to the real application scenario.
>
> **Q3: Whenever not sure about the gender of the referring person, it's better to use THEY as pronoun instead of saying he/she all the time. It also easier to read this way.**
>
> **Response:** Thanks for the suggestion and we have corrected that in the revised paper.
>
> **Q4: Figure 4 x axis should number of likes in plural.**
>
> **Response:** Thanks for the suggestion and we have corrected that in the revised paper.
>
> **Q5: Figure 4 caption can be further improved by stating that the data presented is per user not per video, it can be confusing if one reads the figure before reaching the paragraph in which it is mentioned.**
>
> **Response:** Thanks for the suggestion and we have corrected that in the revised paper.

---

> > ### Comment · Reviewer_wKmC · 2024-11-25
> > **Follow-up to authors**
> >
> > Thank you for your detailed response. I appreciate the effort to address my points. However, I would like to provide additional feedback on the two main issues I raised:
> >
> > - **Dataset Demographic Bias:** While I understand and agree that collecting data across countries poses significant challenges, the dataset's focus on Chinese users does introduce a notable limitation. This demographic specificity means that any conclusions drawn from the data are primarily applicable to Chinese users and should not be generalized without caution. While I recognize the dataset's diversity in other demographic dimensions, I strongly suggest that the paper explicitly acknowledge this limitation. Clearly stating that the dataset is tailored to Chinese users would improve transparency and help future researchers interpret and use the data appropriately.
> >
> > - **Video Categorization:** The explanation provided in your response is appreciated but remains somewhat vague. For instance, the clustering process for Category I—"Videos were first clustered based on similarities in their title and tag texts"—raises several questions. What specific clustering method was used? Was it fully automated, manual, or a combination of both? If manual intervention was involved, what criteria guided this process? Additionally, why were 37 categories chosen? Was this number determined empirically or through a specific rationale? The same questions apply to the formation of Categories II and III. I believe that a more detailed description of the clustering methodology and the decision-making process behind the hierarchical structure is necessary to fully understand and assess the taxonomy of the data being proposed.
> >
> > I believe I have given a fair score to this paper and will keep it as is for now. I do not view the baseline diversity as a significant issue (as other reviewers have pointed out), as the primary contribution of this paper lies in the dataset itself. However, for me to consider raising the score, the presentation of the data must be crystal clear, particularly regarding the points above.

---

> ### Author Response · Authors · 2024-11-25
> **Response to Reviewer wKmC (part 2)**
>
> **Q6: Figure can be moves so they are closer to the place they are mentioned. Right now Figures 3 and 4, are like 2 to 3 pages away from the paragraph they are mentioned in. It would be better if authors manage to arrange them in a way that they are closer to the sections they belong to.**
>
> **Response:** Thanks for the suggestion and we have corrected that in the revised paper.
>
> **Q7: What are the implications on the fact that most of the users from the dataset come from a single country? What are the benefits from it? What are the downsides of having so little diversity in terms of country of origin? Could the conclusions made with this data be extrapolated to other places? If not, what could be the solution for this?**
>
> **Response:** Please see the response to Q1.
>
> **Q8: What was the criteria to pick the classes for Category I?**
>
> **Response:** Videos were first clustered based on similarities in their title and tag texts. Each cluster was manually reviewed to define the overarching theme it represented, leveraging human expertise to finalize the content of Category I. To ensure accuracy, every video was manually checked to confirm that its assigned category matched its actual content, with misclassified videos corrected.
>
> **Q9: What was then the Criteria to make the hierarchies from I to II and III?**
>
> **Response:** It's also based on video clustering and human experience to form these categories, see the response to Q1 for more detail.
>
> **Q10: Although the authors have been very transparent about their data collection and statistics. I still think that a closer look to the data release and the biases on the dataset should be checked. I am no expert in these topics and from my perspective looks like the authors have been transparent and have everything in order for the release of the data. However, I still think that a closer look into it would be beneficial given that the dataset might contain sensitive information about the users involved and their preferences and interactions with the platform.**
>
> **Response:** We would like to further clarify the user information in our dataset. All the participants have signed the informed consent form and agreed their anonymized information to be used for research purposes. For each participant, we collected seven pieces of information: user ID, gender, age, device price, name of the residential city, city's level, and community type. We provide a specific example in the following table. The user ID has been hashed, which was mapped to a number between 1 and 10,000 to prevent the identification of users. For other demographic and geographic attributes, we only collected coarse-grained data, for example, the geographic granularity is limited to the city level. Such information is not sufficient to be utilized for the identification of users.
>
> | User-side information        | Example           |
> | ---------------------------- | ----------------- |
> | user id                      | 631               |
> | gender                       | 'M' (male)        |
> | age                          | 29                |
> | device price                 | 6222              |
> | name of the residential city | 'Shenzhen'        |
> | city's level                 | 'first-tier city' |
> | community type               | 'urban area'      |

---

### Official Review · Reviewer_RoD3 · 2024-11-04

**Soundness:** 3
**Presentation:** 3
**Contribution:** 3
**Rating:** 6
**Confidence:** 4

**Summary:**

In this paper, a large-scale dataset with behavior, attributes, and content of mobile short-video platform is proposed. The proposed dataset includes 10,000 voluntary users and 153,561 videos. The authors perform three-fold technical validations of the dataset, focusing on the richness of behavior data (such as interaction frequency and feedback distribution), the extensive coverage of both user-side and video-side attribute data, and the representational capacity of the content features. This dataset aims to support a wide range of research areas, including but not limited to user modeling, social sciences, and understanding human behavior.

**Strengths:**

1. The paper is well organized and presented, which is easy to follow.
2. The scale of the dataset is large, which can support the training of large models and promote the development of applications in different fields. It is good for the community.
3. The paper provides a detailed introduction of the dataset, including the specific compositions of behavior data, attribute data and content data.
4. The authors conduct statistical analysis of data distributions, interaction numbers, as well as the associations between behaviors and preferences.

**Weaknesses:**

1. The dataset proposed in the paper lacks testing in tasks of different research areas (such as user modeling, social sciences, and understanding human behavior) mentioned in the paper, which results in insufficient clarity regarding its practicality and applicability.
2. Different tasks and methods have not been tested on this dataset, resulting in an incomplete benchmark. This may lead to confusion regarding the use of the dataset due to the absence of reference benchmark results.
3. Visual examples of different data such as video, images, text, and audio of behavior, attribute, and content data should be given. Both the paper and the supplementary material do not provide sufficient visual instances. It is not good for readers.
4. There are some minor grammar errors and typos.

**Questions:**

See weakness.

---

> ### Author Response · Authors · 2024-11-25
> **Response to Reviewer RoD3**
>
> **Q1: The dataset proposed in the paper lacks testing in tasks of different research areas (such as user modeling, social sciences, and understanding human behavior) mentioned in the paper, which results in insufficient clarity regarding its practicality and applicability.**
>
> **Response:**
>
> Thanks for the suggestion. We have supplemented experimental results of 8 recommendation algorithms (including both general and multimodal recommendation methods) to validate the practical usage of our dataset on user modeling. The results are as follows:
>
> |  Model   | Recall@10 | Recall@20 | Recall@50 | NDCG@10 | NDCG@20 | NDCG@50 |
> | :------: | :-------: | :-------: | :-------: | :-----: | :-----: | :-----: |
> |   BPR    |  0.0113   |  0.0218   |  0.0453   | 0.0078  | 0.0114  | 0.0179  |
> | LightGCN |  0.0223   |  0.0358   |  0.0629   | 0.0169  | 0.0211  | 0.0286  |
> | LayerGCN |  0.0208   |  0.0336   |  0.0559   | 0.0160  | 0.0198  | 0.0260  |
> |   VBPR   |  0.0184   |  0.0283   |  0.0519   | 0.0140  | 0.0172  | 0.0237  |
> |  MMGCN   |  0.0105   |  0.0187   |  0.0330   | 0.0088  | 0.0114  | 0.0155  |
> |   GRCN   |  0.0119   |  0.0224   |  0.0444   | 0.0089  | 0.0125  | 0.0189  |
> |   BM3    |  0.0238   |  0.0364   |  0.0638   | 0.0178  | 0.0218  | 0.0294  |
> |  LGMRec  |  0.0159   |  0.0257   |  0.0449   | 0.0131  | 0.0162  | 0.0217  |
>
> As for social science and human behavior understanding research, we provide some preliminary ideas for studying filter bubble phenomenon with our dataset as an example. The filter bubble refers to the situation where recommendation algorithms overly tailor content to a user's preference, resulting in limited content exposure for the user on the platform. To quantify content coverage, our dataset includes a hierarchical content category system. Coverage at a category level c is defined as
> $$
> \frac{N_{seen}(u,c)}{N_{all}(c)}
> $$
> where N_seen represents the number of level-c categories a user has interacted with over a period, and N_all is the total number of level-c categories available. A user's filter bubble extent can then be determined by comparing this metric against a threshold, such as the average across all users. Subsequent analyses could investigate how user behaviors, such as viewing time and content preferences, influence the severity of filter bubbles across different category levels. Additionally, tracking the temporal evolution of filter bubbles at each level could provide insights into their dynamics and persistence over time.
>
> **Q2: Different tasks and methods have not been tested on this dataset, resulting in an incomplete benchmark. This may lead to confusion regarding the use of the dataset due to the absence of reference benchmark results.**
>
> **Response:** We have supplemented experimental results from nine recommendation algorithms to validate the utility of our dataset for user modeling. Additionally, we provided hypothetical use cases and research ideas on studying the filter bubble phenomenon to further demonstrate its practical applicability. For detailed explanations, please refer to the response to Q1.
>
> **Q3: Visual examples of different data such as video, images, text, and audio of behavior, attribute, and content data should be given. Both the paper and the supplementary material do not provide sufficient visual instances. It is not good for readers.**
>
> **Response:** We have supplemented some visual examples of video content in out dataset, including images, titles, tags, ASR texts, categories and duration in Figure 11 of the Appendix. The attribute data for both users and videos has been presented in Table 3 of the Appendix. The raw videos are listed on the website.
>
> **Q4: There are some minor grammar errors and typos**
>
> **Response:** Thanks for pointing out, we have further checked the grammar and typo issue, the paper has been updated.

---

> > ### Comment · Reviewer_RoD3 · 2024-11-27
> > **Reply to authors**
> >
> > I appreciate the efforts to address my comments.
> >
> > Although some experimental results on the proposed dataset are provided, the baseline construction and analysis is still limited, particularly in the following two aspects:
> > 1) Lack of detailed experimental analysis. It is suggested to perform a deeper analysis of the experimental results based on the characteristics of the tested methods and the specific properties of the proposed dataset.
> > 2) It seems that only experimental results for recommendation methods are reported. Does it imply that the dataset is intended only for algorithms in the recommendation field? If not, it is suggested to provide results on other tasks.
> >
> > For a new dataset, it is crucial to establish sufficient baselines to demonstrate the practicality and applicability of the dataset. Other reviewers also indicate the importance of the experiments and analysis. However, the construction and analysis of baselines in this paper are inadequate. Therefore, I maintain my original rating.

---

### Note · Authors · 2024-12-17

I have read and agree with the venue's withdrawal policy on behalf of myself and my co-authors.